

# Top-down estimate of black carbon emissions for city cluster

# using ground observations: A case study in southern Jiangsu,

# China

Xuefen Zhao[1], Yu Zhao[1,2*], Dong Chen[1], Chunyan Li[3], Jie Zhang[3]

1. State Key Laboratory of Pollution Control and Resource Reuse and School of the
Environment, Nanjing University, 163 Xianlin Ave., Nanjing, Jiangsu 210023, China

2. Jiangsu Collaborative Innovation Center of Atmospheric Environment and
Equipment Technology (CICAEET), Nanjing University of Information Science and
Technology, Jiangsu 210044, China

3. Jiangsu Provincial Academy of Environmental Science, 176 North Jiangdong Rd.,
Nanjing, Jiangsu 210036, China

*Corresponding author: Yu Zhao

Phone: 86-25-89680650; email: *yuzhao@nju.edu.cn*



## 20 **Abstract**

We combined a chemistry transport model (CTM), a multiple regression model and
available ground observations, to derive top-down estimate of black carbon (BC)
emissions and to reduce deviations between simulations and observations for southern
Jiangsu city cluster, a typical developed region of eastern China. Scaled from a
high-resolution inventory for 2012 based on changes in activity levels, the BC
emissions in southern Jiangsu were calculated at 27.0 Gg/yr for 2015 (JS-prior). The
annual mean concentration of BC at Xianlin Campus of Nanjing University (NJU, a
suburban site) was simulated at 3.4 $\mu g/m^3$, 11% lower than the observed 3.8 $\mu g/m^3$. In
contrast, it was simulated at 3.4 $\mu g/m^3$ at Jiangsu Provincial Academy of
Environmental Science (PAES, an urban site), 36% higher than the observed 2.5
$\mu g/m^3$. The discrepancies at the two sites implied the uncertainty of the bottom-up
inventory of BC emissions. Assuming a near-linear response of BC concentrations to
emission changes, we applied a multiple regression model to fit the hourly surface
concentrations of BC at the two sites, based on the detailed source contributions to
ambient BC levels from brute-force simulation. Constrained with this top-down
method, BC emissions were estimated at 13.4 Gg/yr (JS-posterior), 50% smaller than
the bottom-up estimate, and stronger seasonal variations were found. Biases between
simulations and observations were reduced for most months at the two sites when
JS-posterior was applied. At PAES, in particular, the simulated annual mean was
elevated to 2.6 $\mu g/m^3$ and the annual normalized mean error (NME) decreased from
72.0% to 57.6%. However, application of JS-posterior slightly enhanced NMEs in
July and October at NJU where simulated concentrations with JS-prior were lower
than observations, implying that reduction in total emissions could not correct CTM
underestimation. The effects of numbers and spatial representativeness of observation
sites on top-down estimate were further quantified. The best CTM performance was
obtained when observations of both sites were used with their difference in spatial
functions considered in emission constraining. Given the limited BC observation data



in the area, therefore, more measurements with better spatiotemporal coverage were
recommended for constraining BC emissions effectively. Top-down estimates derived
from JS-prior and the Multi-resolution Emission Inventory for China (MEIC) were
compared to test the sensitivity of the method to initial emission input. The
differences in emission levels, spatial distributions and CTM performances were
largely reduced after constraining, implying that the impact of initial inventory was
limited on top-down estimate. Sensitivity analysis proved the rationality of near
linearity assumption between emissions and concentrations, and the impact of wet
deposition on the multiple regression model was demonstrated moderate through data
screening based on simulated wet deposition and satellite-derived precipitation.

## 1 Introduction

Black carbon (BC), alternatively referred as elemental carbon (EC), is an crucial
component of atmospheric particle and comes mainly from incomplete combustion of
fossil fuels and biomass. BC has adverse effect on human health as it absorbs harmful
volatile organic compounds like polycyclic aromatic hydrocarbons (Dachs and
Eisenreich, 2000). Furthermore, BC contributes to global warming by intercepting
and absorbing sunlight (Jacobson, 2001; Ramanathan and Carmichael, 2008). Bond et
al. (2013) assessed that the global average radiative forcing of BC was +1.1 $W/m^2$
(90% confidence interval: 0.17-2.1 $W/m^2$), which was more than two-thirds of that
from $CO_2$ (+1.56 $W/m^2$). Since BC remains for only a few days in the atmosphere, it
is an effective way to mitigate climate warming in the short term by reducing BC
emissions. However, due to lack of sufficient understanding of major emission
sources, the effect of BC on regional climate was not fully quantified by models.
BC emission inventories are traditionally developed with the bottom-up method
based on activity levels and emission factors. Previous studies of chemistry transport
modeling (CTM) based on emission inventories found large discrepancies between
simulated and observed BC concentrations. Koch et al. (2009) found that sixteen
models applied in the AeroCom aerosol model inter-comparison project



underestimated surface BC levels by a factor of 2-3. Hu et al. (2016) found that CTM
significantly underestimated the peak surface concentrations of BC over northwestern
United States, likely due to missing strong local fire events in fire emissions.
Moreover, large differences existed in various bottom-up emission inventories,
particularly for China with large energy consumption, complicated emission source
categories, and fast changes in emission characteristics. BC emissions in China for
2001 and 2006 in the Regional Emission inventory in ASia (REAS 2.1, Kurokawa et
al., 2013) were smaller than those in the Intercontinental Chemical Transport
Experiment-Phase B (INTEX-B, Zhang et al., 2009), but the growth rate of BC
emissions in REAS 2.1 was larger than that in INTEX-B (30% versus 15%) for the
five years. Ohara et al. (2007) evaluated the inter-annual trend in China's BC
emissions with constant emission factors, and found that the national emissions
continuously decreased by 23% from 1990 to 2000. In contrast, Lei et al. (2011)
suggested a much smaller inter-annual variability with the peak annual emissions
found in 1996 for the same period. The differences resulted largely from the use of
activity levels from various data sources, especially for residential biofuel combustion.
The gaps between different studies implied potentially large uncertainties in BC
bottom-up emission inventories. The uncertainties of BC emission estimates for China
were reported at ±484%, ±208%, and ±98% by Streets et al. (2003), Zhang et al.
(2009), and Lu et al. (2011), respectively. Due to lack of sufficient local field tests,
emission factors were commonly taken from foreign studies with big variety
depending on fuel and combustion condition (Bond et al., 2004; Cao et al., 2006; Lei
et al., 2011; Qin and Xie, 2012; Streets et al., 2003; Streets et al., 2001; Zhang et al.,
2009). It was also difficult to obtain accurate and detailed activity data, particularly
for the main sources of BC including small industries (e.g., coke and brick
production), off-road transportation, and residential solid fuel combustion. Besides the
large uncertainty in emission estimation, challenges existed as well in updating BC
inventories continuously (Hong et al., 2017; Lu et al., 2011; Xia et al., 2016; Zhao et



al., 2013). To beat severe air pollution, China has been conducting series of measures
in energy conservation and emission control, leading to dramatic changes in energy
structure, emission factors and removal rates of air pollutant control devices (Zhao et
al., 2014). Such changes could be partly tracked by continuous emission monitoring
system (CEMS) that was commonly installed at big industrial enterprises. Large
fractions of BC emissions, however, came from medium and small sources, and their
most recent improvements in manufacturing technologies and emission controls were
relatively difficult to be obtained timely and efficiently.

Given above limitations in bottom-up inventories, different top-down approaches

were applied to evaluate BC emissions. For example, Cohen and Wang (2014)
presented a Kalman filter technique to estimate the global BC emissions based on
satellite-derived radiances and surface concentrations from global and regional
networks. The adjoint-based 4-D variational approach was also applied to constrain
the bottom-up BC emissions at the global or national scales (Zhang et al., 2015; Xu et
al., 2013; Guerrette et al., 2017). A near-linear response of BC concentrations to
emission changes was generally assumed at national (Fu et al., 2012; Kondo et al.,
2011; Wang et al., 2013) and regional scales (Li et al., 2015; Wang et al., 2011), due to
its weak activity in atmospheric chemistry reaction. The ratio of observed to
simulated concentration can be used as a scaling factor to correct BC emissions.
Kondo et al. (2011) made continuous measurement of BC concentrations for a full
year on a remote island in East China Sea. With the data strongly affected by
emissions from China identified and those largely influenced by wet deposition
excluded, they estimated China's annual anthropogenic BC emissions at 1.92 TgC/yr.
Wang et al. (2013) verified this linearity by conducting sensitivity simulation in which
emissions were increased by 50%. After excluding observation data of heavy
pollution and strong precipitation events at five Chinese sites, they calculated China's
annual BC emissions at 1.80 TgC/yr. The results of both studies were close to a
bottom-up estimate at 1.81 TgC/yr by Zhang et al. (2009). Based on observations at



10 Chinese background and rural sites, Fu et al. (2012) applied a multiple regression
model and CTM to quantify China's BC emissions. They calculated the total
emissions at 3.05 TgC/yr, 59% larger than those by Zhang et al. (2009). Using similar
approach, Li et al. (2015) estimated BC emissions to be 34% larger than bottom-up
inventory in Pearl River Delta in south China by Zheng et al. (2012). Park et al. (2003)
used the multiple linear regression to fit the Interagency Monitoring of Protected
Visual Environments (IMPROVE) data and estimated that BC emissions from fossil
fuel and biofuel burning in the United States should be increased by 15%. Combining
a general circulation model simulation and the receptor modeling approach, Verma et
al. (2017) constrained BC emissions over India based on the scaling factor (the ratio
of simulated to observed BC concentration).

To our knowledge, limitations remained in the assessment of BC emissions based

on the top-down approach. Current available studies focused mainly on global or
national scale, and few evaluations could be found for city clusters. In aims of
examining emission control policies and quantifying impacts of BC on local climate
and air quality, there was a strong need for studies at city cluster scale that require
ground observation and emission inventory with improved details. Regarding
measurement data, monthly or annual means were commonly used in previous studies,
and information of heavy-polluted events were lost when targeting a local scale. In
general, observations at a higher temporal resolution were considered as an important
means to effectively reduce uncertainties (Matsui et al., 2013; Wang et al., 2013;
Gilardoni et al., 2011). Moreover, it was somewhat arbitrary to differentiate emissions
by sector in previous top-down estimates, attributed to lack of detailed information on
source categories from bottom-up inventories. The method was thus insufficient to
make substantial improvement on emission evaluation by sector, or to clearly stress
the direction of further revisions on bottom-up inventories.

In this work, therefore, we integrated CTM, multiple regression model and

available hourly ground observations to provide top-down constraint of BC emissions



and to reduce deviations between simulations and observations at city cluster scale.
We selected southern Jiangsu city cluster including cities of Suzhou, Wuxi,
Changzhou, Zhenjiang, and Nanjing, a typical region with large population and
economy in Yangtze River Delta (YRD), China (see the geographic location and cities
in Figure S1 in the supplement). Given its intensive industry and energy consumption,
the city cluster was regarded as one of the largest BC emission sources in eastern
China and BC emissions from this region accounted for nearly half of the total
emissions in Jiangsu (Zhou et al., 2017). The heavy air pollution was found in the
region: the annual averages of fine particle (PM$_{2.5}$) concentrations in all the cities
exceeded the National Ambient Air Quality Standard (NAAQS, 35 μg/m$^3$) in 2012.
Under the pressure of air quality improvement, Jiangsu conducted aggressive actions
of emission control, leading to 20% reduction in the annual average of PM$_{2.5}$
concentration from 2013 to 2015. Based on a provincial bottom-up emission inventory,
we estimated the contributions to BC concentrations by sector at two ground
observation sites through the brute-force method in CTM. The results, together with
observed ambient BC concentrations, were incorporated in a multiple regression
model to derive the top-down estimate of BC emissions for southern Jiangsu city
cluster. The advantage of top-down estimate against bottom-up inventory was then
judged by CTM and ground observations. The factors that would potentially influence
the top-down estimate were also evaluated, including number and spatial
representativeness of observation sites, and initial bottom-up emission input. The
uncertainties of the multiple regression model were finally evaluated including the
influence of precipitation and the near-linear assumption between BC emissions and
concentrations.

## 2 Data and method

### 2.1 Bottom-up inventories of BC emissions

Two bottom-up emission inventories at different spatial scales were used in this





work. At the national scale, the Multi-resolution Emission Inventory for China (MEIC,
http://www.meicmodel.org/) was developed by Tsinghua University, with an original
horizontal resolution at 0.25°×0.25°. At the provincial scale, Zhou et al. (2017)
collected the best available information of industrial sources in Jiangsu and developed
an inventory with higher resolution at 3×3 km. The latter was proved to be more
supportive in air quality simulation at city cluster scale (Zhou et al., 2017; Zhao et al.,
2017). In both inventories, anthropogenic BC emissions for 2012 came from four
major sectors: power generation, industry, residential sources and transportation. The
national and provincial inventories for 2015 (mentioned respectively as MEIC-prior
and JS-prior hereinafter) were obtained using a simple scaling method based mainly
on changes in activity levels (energy consumption and industrial production, etc)
between the four years. Table S1 in the supplement summarizes the data sources of
activity levels and the scaling factors by sector in JS-prior. As MEIC-prior includes
only four major sectors, the scaling factor for each sector was calculated as the
average of those for subcategories within the sector. Potential changes in BC emission
factors from 2012 to 2015, e.g., those attributed to varied manufacturing technologies
and/or penetrations of emission control devices, were not considered in the calculation.
The implication and uncertainty from that simplified emission scaling method will be
further discussed in Section 4.3. The temporal distribution of the emissions was
dependent on that of activity levels by source category. Such information was
investigated by Zhou et al. (2017) according to the official statistics of the country
(http://data.stats.gov.cn/) and directly adopted in this work.
**2.2 Top-down emission estimation with multiple regression model**
The top-down emissions of BC in southern Jiangsu (mentioned as JS-posterior
hereinafter) were estimated with a multiple regression model using ground
observations as constraint. The regression model matched BC contributions by sector
(calculated through CTM) against measured ambient hourly BC concentrations:





$$c_{obs} = \beta_1 c_{power} + \beta_2 c_{industry} + \beta_3 c_{residential} + \beta_4 c_{transportation} + \varepsilon \qquad (1)$$
where $c_{obs}$ is the vector of observed hourly BC concentrations; $c_{power}$, $c_{industry}$, $c_{residential}$,
and $c_{transportation}$ are the vectors of BC concentrations contributed by power generation,
industry, residential sources and transportation, respectively, and they were simulated
using the brute-force method as described in Section 2.3; $\beta_1$-$\beta_4$ are the domain-wide
scaling factors obtained by sector in the multiple regression model to best match
observations; and $\varepsilon$ is the error vector of the model.
As BC is not one of the six regulated air pollutants in the NAAQS, it was a big
challenge to obtain observation data with high temporal resolution in most cities of
southern Jiangsu. For the whole year 2015, hourly ambient BC concentrations were
available at two sites in Nanjing, the capital of Jiangsu. As illustrated in Figure 1, one
is a suburban site located in the Xianlin Campus of Nanjing University in northeast
Nanjing (NJU), and the other is an urban site in Jiangsu Provincial Academy of
Environmental Science (PAES). At both sites, BC was sampled and analyzed hourly
with semi-continuous carbon analyzer (Model-4, Sunset Lab, USA). Details of the
measurement approach were described in Chen et al. (2017). The statistics of
observed ambient BC concentrations at the two sites are shown in Figure S2 in the
supplement. The annual average BC concentrations (calculated as the mean of January,
April, July and October) were 3.83 and 2.47 $\mu g/m^3$ at NJU and PAES, respectively.
The hourly average BC observations ranged 0.06-17.65 $\mu g/m^3$ and 0.22-19.76 $\mu g/m^3$
at NJU and PAES, respectively. The values were similar to those observed in the
Guanzhong basin (0.4-23.1$\mu g/m^3$), the Pearl River Delta region (1-13 $\mu g/m^3$) and the
Beijing-Tianjin-Hebei region (2-32 $\mu g/m^3$) (Li et al., 2016). Much higher BC
concentrations were observed in autumn and winter at both sites, with the monthly
means at 3.96 and 5.44 $\mu g/m^3$ at NJU and 3.62 and 2.80 $\mu g/m^3$ at PAES, respectively.
The scaling factors derived from Eq. (1) were used to constrain BC emissions in
JS-prior from a top-down perspective by assuming a near-linear relation between
changes in BC concentrations and emissions:




$$E_{JS\text{-}posterior} = \beta_1 E_{power} + \beta_2 E_{industry} + \beta_3 E_{residential} + \beta_4 E_{transportation} \qquad (2)$$
where $E_{JS\text{-}posterior}$ is the vector of the total BC emissions from the top-down approach;
$E_{power}$, $E_{industry}$, $E_{residential}$ and $E_{transportation}$ are the vectors of BC emissions from power
generation, industry, residential sources and transportation, respectively, in JS-prior.

**2.3 Air quality simulation**

We used the Models-3 Community Multi-scale Air Quality (CMAQ) version
4.7.1 to simulate ambient BC concentrations. As shown in Figure 1, three nested
domains were applied with horizontal resolutions of 27, 9, and 3 km, respectively, on
a Lambert Conformal Conic projection centered at (110°E, 34°N). The mother domain
(D1, 177×127 cells) covered most parts of China and other surrounding countries. The
second domain (D2, 118×121 cells) covered Jiangsu, Anhui, Zhejiang, Shanghai, and
parts of other provinces in China . The third domain (D3, 133×73 cells) covered
Shanghai, part of Anhui province and the city cluster in southern Jiangsu. There were
27 vertical levels from the ground surface up to 50 hPa on terrain-following
coordinated. The simulations were conducted for January, April, July and October to
represent four typical seasons in 2015. A 5-day spin-up period of each month was
applied to minimize the influence of initial conditions in the simulations.
Meteorological fields were simulated by the Weather Research and Forecasting
Model (WRF) version 3.4 and the carbon bond gas-phase mechanism (CB05) and
AERO5 aerosol module were adopted in CMAQ. Relevant details of model
configuration can be found in Zhou et al. (2017). Statistical indicators including
averages of simulations and observations, bias, normalized mean bias (NMB),
normalized mean error (NME), root mean squared error (RMSE) and index of
agreement (IOA) were applied to evaluate the modeling performance of WRF (Baker
et al, 2004; Zhang et al., 2006). Ground observation data at 1 or 3 h interval at
meteorological stations including Lukou, Hongqiao and Liyang stations in the third
domain (labeled in Figure 1) were taken from National Climatic Data Center (NCDC).



The statistical indicators for temperature at 2 m (T2) and relative humidity at 2 m
(RH2), wind speed and direction at 10 m (WS10 and WD10) for the four typical
months in 2015 are summarized in Table S2 in the supplement. Discrepancies
between ground observations and WRF modeling were within acceptable range
(Emery et al., 2001).

To make it applicable in our CTM, MEIC-prior was downscaled into grid

systems of each modeling domain, based on the spatial distributions of gross domestic
product (GDP, for power generation and industrial emissions) and population (for
residential and transportation emissions) at a horizontal resolution of 1×1 km. The
downscaled MEIC-prior was used for the first, the second domains and the regions
outside Jiangsu of the third domains, while JS-prior was applied for the Jiangsu region
of the third domain. Brute-force method was applied to estimate contributions to
ambient BC concentrations by sector. Five scenarios were designed in this study:
Scenario B (the base scenario) in which emissions from all sources in the third
domain were included, and Scenarios S1, S2, S3, and S4 in which BC emissions from
power generation, industry, residential sources and transportation were zeroed out,
respectively. We compared simulated BC concentrations in S1, S2, S3 and S4 with
those in Scenario B in four months, and the contributions from four major emission
sectors to ambient BC levels were determined as the differences in simulated
concentrations between Scenarios B and S.
**3 Results**
**3.1 Bottom-up emission estimate**

The total annual BC emissions of JS-prior were estimated at 26.99 Gg for

southern Jiangsu city cluster in 2015, including 0.18 Gg from power generation, 17.67
Gg from industry, 3.80 Gg from residential sources and 5.33 Gg from transportation,
as shown in Figure 2. Accounting for 66% of total annual emissions, industry was
identified as the dominant contributor to BC, followed by transportation (20%) and





residential sources (14%). Although the policies of energy conservation and emission
control have been conducted for years, there were still a number of small facilities
with low operation temperatures and combustion efficiencies in southern Jiangsu,
leading to a large amount of BC from incomplete combustion. When scaling
emissions from 2012 to 2015, in addition, improvements in emission controls were
not taken into account, such as elevated combustion technologies and enhanced use of
dust collectors. The potential reductions in net emission factors for major factories,
therefore, were not well quantified, and the emissions from industry could be
overestimated. Emissions from power generation were few, resulting from relatively
high combustion efficiency of pulverized boilers and large penetrations and removal
rates of dust collectors. Besides the annual total, the emissions of four months
(January, April, July and October) were also estimated and limited seasonal
differences were found as shown in Figure 2.
Figure S3 in the supplement shows the spatial distribution of annual BC
emissions in JS-prior. For power generation and industry sectors, latitude and
longitude of each plant were applied to allocate BC emissions, and the outstandingly
high emissions shown in the map indicated the existence of big industrial plants. For
residential sources, large emissions were found in the regions with intensive
population. Emissions from transportation were mainly distributed along the road net
and downtown regions in southern Jiangsu cities (see the geographic locations of
downtowns in Figure S1 in the supplement), slightly overlapping with those from
residential sources.
**3.2 Top-down emission estimate**
The time series of BC concentrations contributed by various sectors ($c$ in Eq. (1))
were simulated with CTM and illustrated in Figures S4 and S5 in the supplement for
NJU and PAES, respectively. Among all the sectors, the largest seasonal variation in
BC contribution was found for residential sources. The average concentrations
contributed by this sector in January reached 0.76 and 0.94 $\mu g/m^3$ at NJU and PAES,



respectively, approximately double of those in another three months. The
concentrations contributed by industry were significantly enhanced in certain periods
(e.g., January 20[th], April 9[th]-11[th], and July 15[th]-17[th]), and industrial emissions were
expected to be an important reason for the overestimation in BC concentrations
through CTM (see the model evaluation in Section 3.3). Table S3 in the supplement
summarizes the monthly and annual mean BC contributions by sector. The annual
contributions of industry at the two sites were close to each other (21.0% and 21.9%
at NJU and PAES respectively). Contributions of residential sources and
transportation were higher at PAES resulting from large population and heavy traffic
in the urban area. Minor contribution of power generation to BC concentrations was
found at both sites (the annual means were less than 1%), attributed to its very limited
emissions.
Summarized in Table 1 are the scaling factors $\beta_1$-$\beta_4$ estimated from multiple
regression model (Eq. (1)) by season, together with the statistical indicators including
the values of t, Sig. (or p) and variance inflation factor (VIF). The values of t and Sig.
indicate statistical significance with a threshold of 2 and 0.05, respectively. VIF is a
test for multicollinearity and the model is reasonable with VIF smaller than 10. Since
the emissions from power generation were small and they contributed very little to
ambient BC concentrations, inclusion of power generation component would not
significantly improve the regression model. In this study, therefore, we assumed that
the simulated BC concentrations from power generation were correct by setting $\beta_1$ at
1 and further subtracted them from the observations. Most statistical indicators in
Table 1 met the criteria (t>2, Sig.<0.05, VIF<10) and the overall significance was
0.00 in four months, implying acceptable robustness of the multiple regression model.
However, the results were not statistically significant indicated by t and p values for
some months and sectors (e.g., industry in April and residential in April and July),
implying that the constrained emissions for those month/sectors need to be cautiously
analyzed.



By applying $\beta_1$-$\beta_4$ in Eq. (2), the top-down estimates of BC emissions
(JS-posterior) were estimated and illustrated in Figure 2. The total BC emissions for
southern Jiangsu city cluster were calculated at 13.4 Gg, 50% smaller than those of
JS-prior. The scaling factors of emissions from industry and transportation ($\beta_2$ and $\beta_4$)
ranged from 0.22 to 0.42 and from 0.55 to 0.79 for different months, respectively.
Accordingly, the emissions from industry and transportation in JS-posterior were
estimated 67% and 32% smaller than those in JS-prior, respectively. As mentioned
above, the emissions in JS-prior 2015 were simply scaled from those in 2012
according to activity data, and changes in emission factors were not considered. In the
actual fact, however, a series of measures in industry and transportation were
conducted to improve energy efficiency and to reduce emissions over recent years.
Issued in 2013, for example, the Air Pollution Control Planning for the Key Regions
for the 12$^{th}$ Five-Year Plan period (2010-2015) aimed to achieve 7% and 15%
reductions in the annual average concentration and industrial emissions of fine
particles in Jiangsu province from 2010 to 2015, respectively (Qian, 2013). The
measures included eliminating old and energy-inefficient plants of heavy-polluted
industries (thermal power generation and steel/building material production), and
optimizing the energy structure through application of sustainable energy. Meanwhile,
the enhanced use of cleaner gasoline and diesel products (National stage V standard)
in transportation could lead to reduced vehicle emissions. The government efforts in
emissions controls proved effective, indicated by the scaling factors much smaller
than 1 ($\beta_2$ and $\beta_4$ in Table 1) and the reduced emissions of JS-posterior. For residential
sources, the emissions in JS-posterior were 3% smaller than those in JS-prior,
indicating limited difference in the annual total emissions between the two inventories.
However, the scaling factors ($\beta_3$) in January and October were 1.31 and 1.52
respectively, showing a stronger enhancement in BC emissions in winter and autumn
in JS-posterior than those in JS-prior. It thus implied that there were missing sources
likely associated with low-quality fossil fuels or biofuel used for heating in winter and
crop waste burning in autumn in JS-prior. For the capital city of Jiangsu Province,
Nanjing, Huang et al. (in preparation) conducted detailed analysis on the changes in
operation activities and emission control technologies of individual sources based on
annually updated official environmental statistics and pollution census. With the
bottom-up approach, the annual BC emissions in the city were estimated to decrease
by 60% from 2012 to 2015 as shown in Figure S6 in the supplement. The relative
change in annual emissions was close to that between JS-prior and JS-posterior, and
the validity of the two methods (the bottom-up approach by Huang et al. and the
top-down approach in this work) could be verified.

Figure 3 presents the seasonal variations in BC emissions of JS-prior,

JS-posterior and MEIC-prior by sector, and stronger variations were generally found
in JS-posterior. As shown in Figure 3a, the largest difference among the three
inventories existed in the residential sources, and the ratio of maximum to minimum
monthly emissions was 4.33 in JS-posterior, close to that in MEIC-prior at 4.00 and
nearly 4 times of that in JS-prior at 1.13. The analogue ratio for industry was 2.05 in
JS-posterior, nearly twice of those in JS-prior at 1.14 and MEIC-prior at 1.12. The
smallest difference was found for transportation among the three inventories.
Seasonal variations in total emissions were a combination of those by sector weighted
by the contribution of each sector to total emissions. The ratios of maximum to
minimum monthly emissions were 1.13, 1.83 and 1.29 for JS-prior, JS-posterior and
MEIC-prior, respectively (Figure 3b). The value for JS-posterior was closer to 2.1 for
an anthropogenic BC emission inventory in China by Lu et al. (2011) that considered
enhanced use of fossil fuels for residential heating in winter in northern China. The
comparison thus implied again that current bottom-up inventories might
underestimate the emissions of residential solid fuel burning in winter in southern
Jiangsu. As central household heating was not conducted in the area in winter, the
official energy statistics on which bottom-up inventories were based may not fully
capture the elevated fuel burning by disperse households. Spatial distribution of BC



emissions in JS-posterior was illustrated in Figure S3 in the supplement. Compared to
JS-prior, BC emissions from industry and transportation were greatly reduced in
downtown regions in southern Jiangsu city cluster.

**3.3 Evaluation of the top-down emission estimate**

The simulated BC concentrations based on bottom-up (JS-prior) and top-down
estimation in emissions (JS-posterior) were compared with observations to evaluate
the two inventories, and the results were illustrated in Figures 4 and 5 for NJU and
PAES sites, respectively. Statistical indicators including mean concentrations from
simulations and observations, NMB and NME, as well as the regression correlation (R)
were calculated to evaluate the modeling performance, as summarized in Table 2.
In general, CTM based on JS-prior reproduced well the temporal variations of
the observed BC concentrations at the two sites. The highest and lowest
concentrations were respectively simulated in winter and summer, consistent with
observations with an exception at PAES where the observed monthly mean in January
(2.80 $\mu g/m^3$) was lower than that in October (3.62 $\mu g/m^3$). The seasonal variation of
BC concentrations at NJU was larger than that at PAES, suggesting bigger impact of
household solid fuel use on the suburban and rural regions. Though the model was
able to capture the seasonal variability, discrepancies between simulations and
observations existed, and CTM commonly underestimated BC concentrations at the
suburban site NJU and overestimated those at the urban site PAES. With the monthly
means ranged 1.99-5.97 $\mu g/m^3$ at NJU, the annual average of BC concentration
(calculated as the mean of January, April, July and October) was simulated at 3.44
$\mu g/m^3$, smaller than the observed 3.83 $\mu g/m^3$. With the monthly means ranged
2.61-6.46 $\mu g/m^3$, in contrast, the annual concentration at PAES was simulated at 3.39
$\mu g/m^3$, larger than the observed 2.48 $\mu g/m^3$. Better correlation between observation
and simulation was found at NJU, indicated by the larger R. The annual mean NMBs
were calculated at -10.16% and 36.67%, and the NMEs were 41.15% and 72.00% at
NJU and PAES, respectively. The discrepancy suggested that JS-prior used in CTM



might misrepresent the spatial pattern of emissions. Population and economy densities
were applied to allocate BC emissions, leading to overestimation in emissions and
thereby simulated concentrations in urban areas with more population and economic
activity. Besides, the model overestimated the peak surface concentrations at both
sites particularly when the contribution from industry sector was enhanced as
mentioned in Section 3.2 (e.g., January 9th-11th and April 9th-10th at NJU, and April
9th-12th, the second half of July, and October 20th at PAES).

Application of JS-posterior in CTM effectively corrected large biases between

simulations and observations at the two sites. As shown in Table 2, NMEs were
reduced for most months while effects of applying JS-posterior in CTM varied at two
sites. At PAES, the annual average NME declined from 72.00% to 57.55% and the
annual mean of BC concentration was simulated at 2.57 μg/m$^3$, in better agreement
with the observed 2.48 μg/m$^3$ than the simulated 3.39 μg/m$^3$ using JS-prior. The
largest reductions in NMEs were found in April and July, from 73.18% to 42.87% and
from 92.74% to 42.37%, respectively. Moreover the overestimation in peak
concentrations using JS-prior were partly corrected when JS-posterior was applied,
resulting mainly from the reduced emissions from industry and transportation.
Although simulation of peak concentrations at NJU were improved as well, the annual
average NME at NJU slightly increased from 41.15% to 44.16% and the annual mean
of BC concentration was simulated at 2.82 μg/m$^3$, smaller than the simulated 3.44
μg/m$^3$ using JS-prior. Bigger bias was found in July and October at NJU, since the
reduced emission estimates in JS-posterior led to further underestimation in simulated
ambient BC levels compared to JS-prior. Limitation of current multiple regression
model was thus indicated that overestimation and underestimation in concentrations at
different sites could hardly be corrected simultaneously without further improvement
in spatial distribution of emissions.



## 4 Discussions


We selected April to evaluate the sensitivity of observation and bottom-up

emission input to top-down constraint. Observation site number, spatial
representativeness of sites, and initial bottom-up inventory were changed separately in
the constraining approach, and various top-down estimates could be derived and
compared with each other. The statistical indicators of modeling performances based
on different bottom-up and top-down emission estimates in April are summarized in
Table 3. Furthermore, we evaluated the uncertainty of the multiple regression model,
including the assumption of near linearity between emissions and concentrations and
the impact of precipitation. Details were described as below.

### 4.1 The effect of observation site number


A major challenge in understanding the sources and distributions of BC in China

was lack of a consistent and stable measurement network with good spatiotemporal
coverage, such as the IMPROVE network in the United States (Malm et al., 1994).
Uncertainty existed in the top-down estimates in this work, as hourly measurements
on BC concentrations were only available at two sites in southern Jiangsu. Therefore,
besides JS-posterior derived from observations at both sites as described in Section
3.2 (mentioned as Case 1 hereinafter), we conducted a Case 2 in which observation
data at only one site (NJU) was used in the top-down approach, to analyze the effect
of the site number on emission estimates. The scaling factors of emissions from
industry, residential sources and transportation were recalculated at 0.42, 0.95 and
0.65, respectively. Compared with Scenario B, the NMEs of Case 2 decreased from
42.31% to 32.47% and from 73.18% to 61.59% at NJU and PAES, respectively,
implying the benefits of ground measurements (even available only at one site) on
emission constraint. The NME in Case 2 was slightly smaller than that in Case 1 at
NJU, suggesting that application of measurement data at one single site could
improve model performance moderately at that site. At PAES, in contrast, much larger



NME was found in Case 2. Much better model performance in Case 1 at PAES
indicated that inclusion of more measurements with better spatiotemporal coverage
could constrain BC emissions at city cluster level more effectively.
**4.2 The effect of spatial representativeness of observation sites**

Spatial representativeness of observation sites was identified and its impact on

top-down emission constraint was evaluated. Considering the prevailing winds from
northeast and southeast, on one hand, NJU located upwind Nanjing is hardly
influenced by the emissions from the downtown of the city. Besides the site is
downwind of the Yangtze River Delta region (YRD) including the
Suzhou–Wuxi–Changzhou-Zhenjiang city cluster (Chen et al., 2017), thus it is more
representative for the western YRD emissions through regional transport. On the other
hand, PAES is located at urban Nanjing and its air quality is commonly influenced by
surrounding transportation, residential, and commercial sources, thus the site is
representative for the local emissions of Nanjing. In contrast to previous top-down
studies that did not distinguish influence of local emissions and transport on air
quality in sub-regions of the research domain (Wang et al., 2011; Fu et al., 2012), the
spatial representativeness of the two observation sites were taken into account to
improve the top-down approach and the result of constraining BC emissions in
southern Jiangsu city cluster. Through the brute-force method described in Section 2.3,
we zeroed out the emissions from Nanjing and Suzhou–Wuxi–Changzhou-Zhenjiang
city cluster in CTM, respectively, and compared the simulated concentrations with
those in Scenario B to analyze the contributions of the two regions to ambient BC
concentrations at NJU and PAES sites. As shown in Figure S7 in the supplement, the
contribution of emissions from Nanjing to PAES was greater than that to NJU in 82%
of the modeling period, and the analogue number was 81% for the contribution of
Suzhou–Wuxi–Changzhou-Zhenjiang city cluster to NJU greater than that to PAES.
We thus concluded that emissions from Nanjing contributed significantly to PAES
while those from Suzhou–Wuxi–Changzhou-Zhenjiang city cluster contributed



significantly to NJU. We then developed a new case of top-down emission estimate in
southern Jiangsu (Case 3), in which observation data at PAES and NJU were applied
to constrain emissions from Nanjing and Suzhou–Wuxi–Changzhou-Zhenjiang city
cluster, respectively.

The scaling factors in Case 3 are provided in Table 4. To avoid the collinearity in

the multiple regression model, we expected that the relative changes in emissions
from transportation in Nanjing and Suzhou–Wuxi–Changzhou-Zhenjiang city cluster
were similar for recent years, resulting from the same progress of emission standard
implementation (National Standard Stage IV) in southern Jiangsu and the frequent
circulation of vehicles among the cities. Therefore a same scaling factor was assumed
for transportation in the two regions. As shown in Table 4, all the scaling factors at
PAES were smaller than those at NJU, implying that implementation of emission
controls    in    Nanjing    were    more    stringent    than    that    in
Suzhou–Wuxi–Changzhou-Zhenjiang city cluster from 2012 to 2015. As the host city
of the 2$^{nd}$ Asian Youth Games in 2013 and the 2$^{nd}$ Youth Olympic Games in 2014,
Nanjing was undertaking series of restrictions on air pollutant emissions. The city
conducted emission control action on small coal-fired boilers since 2013 and over
1200 coal-fired boilers had been shut down by the end of 2014. In addition, central
heating units were largely applied to replace the coal with electricity, natural gas or
biofuel. As shown in Table 3, the NMEs in Case 3 were the smallest at both sites
among all the cases with an exception: the NME at NJU in Case 3 was 32.64%,
slightly larger than that in Case 2 at 32.47%. The result implied that inclusion of more
measurement data with their spatial representativeness considered could improve the
top-down approach in terms of spatial distribution of emissions and could reduce the
deviation between observations and simulations.

Summarized    in    Table    5    are    BC    emissions    from    Nanjing    and

Suzhou–Wuxi–Changzhou-Zhenjiang city cluster estimated in different cases. All the
top-down estimates were approximately half of the bottom-up estimate and the



estimate in Case 1 was the smallest among all the cases. The same scaling factors
were generated and applied in Cases 2 and 3 to calculate BC emissions from
Suzhou–Wuxi–Changzhou-Zhenjiang city cluster which accounted for 80% of the
total emissions in southern Jiangsu, resulting in similar top-down emission estimates
between the two cases.
**4.3 The effect of initial bottom-up emission input**
Given the large uncertainty in JS-prior that was simply developed based on the
changes of activity levels in recent years, we applied MEIC-prior as well to explore
the effect of initial emission inventory on top-down BC constraints.
Figures 6 and 7a compare the total amount and spatial distribution of emissions
between JS-prior and MEIC-prior in April, respectively. The total BC emissions of
southern Jiangsu city cluster in JS-prior were 21% lower than those in MEIC-prior. In
JS-prior, as shown in Figure 7a, the emissions from some industrial plants were
extremely larger than those in MEIC-prior, while the emissions in urban areas were
found smaller. Both inventories indicated extremely small contribution from power
generation. BC emissions from industry sector were calculated at 1.34 Gg in JS-prior,
0.22 Gg smaller than MEIC-prior. Emissions from industry in MEIC-prior were
calculated based on regional average of emission factors and allocated according to
spatial distribution of GDP. The method would possibly result in underestimation in
emissions from big industrial plants but overestimation in urban areas. Emissions
from residential sources in JS-prior were close to those in MEIC-prior as similar
methodology was applied for the sector in the two inventories. BC emissions from
transportation in MEIC-prior (0.85 Gg) were twice of those in JS-prior (0.42 Gg)
attributable probably to the application of different emission factors. For on-road
transportation, the emission factors in JS-prior were calculated with CORPERT model
(EEA, 2012; Zhou et al., 2017) while they were obtained from available domestic
measurements in MEIC-prior.
Simulation Case 4 was determined using MEIC-prior in CTM. As shown in



Table 3, the hourly average of BC concentrations at NJU was simulated at 2.49 μg/m$^3$
for April 2015 in Case 4, close to 2.38 μg/m$^3$ simulated with JS-prior (Scenario B). At
PAES, however, application of MEIC-prior in CTM resulted in much larger
concentration than JS-prior (5.13 versus 2.98 μg/m$^3$), indicating again that
MEIC-prior would overestimate the emissions in urban area. Following the top-down
approach described in Section 2.2, we developed Case 5, using MEIC-prior instead of
JS-prior as the initial input of emission data in CTM. The scaling factors of emissions
from industry, residential sources and transportation were respectively calculated at
0.15, 1.30 and 0.25 through multiple regression model, and the top-down estimate in
BC emissions (mentioned as MEIC-posterior hereafter) were calculated at 0.75 Gg in
April 2015, close to 0.78 Gg in the JS-posterior (Figure 6). The differences in the
emissions from industry and transportation between JS-posterior and MEIC-posterior
were 0.06 and 0.07 Gg, respectively, much smaller than those between JS-prior and
MEIC-prior. Besides the total amount, differences in spatial distribution in industry
plants and urban area between the top-down estimates (JS-posterior and
MEIC-posterior) were also significantly reduced compared to those between
bottom-up estimates (JS-prior and MEIC-prior), as shown in Figure 7b. Figure 8
illustrates the scatterplots of the simulated BC concentrations from bottom-up and
top-down inventories at NJU (Figure 8a) and PAES (Figure 8b). Using two bottom-up
inventories in CTM, bigger difference in simulated BC concentrations was found at
PAES compared to that at NJU, indicated by the slope (1.10) closer to 1 at NJU in
Figure 8a. The correlation coefficients (R$^2$) between simulated BC concentrations
using JS-prior and MEIC-prior were 0.81 at NJU and 0.40 at PAES respectively.
Using two top-down estimates, the difference between simulated concentrations at
PAES was significantly reduced and the slope got much closer to 1 in Figure 8b. The
correlation coefficients (R$^2$) were enhanced to 0.94 and 0.87 at NJU and PAES,
respectively. To summarize, similar results from top-down constraint approach could
be obtained in emission level, spatial distribution, and CTM performance, even clear





difference existed in the initial bottom-up inventories. In other word, limited effect of
initial emission input was evaluated on the top-down estimate from the multiple
regression model.
**4.4 Uncertainty analysis of the multiple regression model**

As mentioned in Section 2.2, the assumption of near linearity between emissions

and concentrations is a principle of the multiple regression model, given the weak
chemistry reactivity of BC. The principle has been applied in previous studies to
constrain BC emissions (Fu et al., 2012; Kondo et al., 2011; Wang et al., 2013; Park et
al., 2003; Verma et al., 2017). In the actual fact, however, processes other than
chemical reaction, e.g., precipitation or wet deposition, impact the linearity. Therefore,
the near-linear assumption needs to be justified, and the uncertainty of the
methodology could then be evaluated.

Sensitivity analysis was conducted to assess the rationality of brute-force method

described in Section 2.3, in which emissions of given sector were zeroed out to
determine their contribution to the ambient concentrations. As summarized in Table
S4 in the supplement, we first calculated the ratio of simulated wet deposition to
emissions by month for NJU, PAES and the whole southern Jiangsu city cluster with
JS-prior (Scenario B) and JS-posterior (Case 1), respectively. July and October were
identified as the months with the most and least impact from precipitation, suggested
by the largest and smallest ratio, respectively. Two sensitivity simulations were then
conducted for the selected two months, in which doubled and halved emissions (i.e.,
200% and 50% of emissions in JS-prior, respectively) were used in CTM, and the
simulated concentrations were then compared to those with JS-prior (i.e., Scenario B).
Figures 9 and 10 illustrate the linear correlations of the simulated concentrations in
these two sensitivity cases and the base scenario (Scenario B) at NJU and PAES,
respectively. As can be seen in all the panels, the fraction of change in simulated
monthly average concentration ($F_{conc.}$) was close to that of emission change ($F_{emis.}$),
i.e., the ratio of $F_{emis.}$ to $F_{conc.}$ was around 1.0, within a range of ±10%. Similar ratio of



change in emissions ($\triangle E$) to that in simulated average concentration ($\triangle C$) was
obtained for each month and site as well. The results thus suggested that the impact of
non-linearity between emissions and concentrations was limited, no matter the
precipitation was strong or not. As the top-down constrained emissions (JS-posterior)
were 50% smaller than the bottom-up estimates (JS-prior), the relative change was far
beyond the uncertainty from non-linearity (±10%), implying the improvement of the
top-down approach on emission estimation.
Many studies have reported the difficulty in precipitation simulation with WRF
(Annor et al., 2017; Liu et al., 2018; Yu et al., 2011; Yang et al., 2014; Kaewmesri,
2018). In this study, the observed ground precipitation at Lukou, Liyang and Shanghai
stations (see Figure 1 for locations) was compared with the simulated one to evaluate
the WRF performance for precipitation modeling. As shown in Figures S8-11 in the
supplement, the model could capture the dates of precipitation, but it generally
overestimated the amount. Similar results were found in previous studies that WRF
overestimated precipitation at fine spatial resolution (Politi et al., 2018; Kotlarski et
al., 2014; García-Díez et al., 2015). Improvement in physics parameterization
schemes in WRF will help better understanding the wet deposition of BC through
simulation. To further evaluate the effect of wet deposition on emission constraining,
we conducted an extra Case 6, in which the data influenced by simulated wet
deposition (i.e., the periods with simulated wet deposition at hourly basis) were
excluded in the top-down approach. The new scaling factors $\beta_1{'}$-$\beta_4{'}$ estimated from the
multiple regression model were summarized in Table 6. By applying $\beta_1{'}$-$\beta_4{'}$ in Eq. (2),
the top-down estimates of BC emissions in Case 6 were calculated at 13.7 Gg, and the
emissions by sector and month were illustrated in Table 7, together with the relative
deviation (RD) compared to emissions in Case 1(JS-posterior). The relative deviations
of monthly total emissions between Case 6 and Case 1 were less than 5%, with an
exception of July at 14%, and that for annual total was 2.6%. Larger relative
deviations were found for given sources, e.g., residential in January and transportation



658 in July. The deviations, therefore, were much smaller than that between the emissions

659 in JS-prior and JS-posterior. We consequently applied CTM to evaluate the model

660 performance with the emissions in Case 6 for July. Illustrated in Table 8 were the

661 simulated BC concentrations and the statistic indicators obtained through comparisons

662 with observation at the two sites. As suggested by the NME and R values, little

663 improvement on CTM performance was achieved with the emissions in Case 6,

664 compared to those with Case 1 (Table 2). The impact of simulated wet deposition on

665 the top-down approach was thus expected to be moderate in this work.

666  As the simulated wet deposition varied from the reality to some extent and the

667 impact of precipitation along the transport was not excluded in Case 6, we selected

668 July to conduct a Case 7, in which the data influenced by accumulative precipitation

669 along the back trajectories at the two sites were excluded in the multiple regression

670 model. The merged high-quality precipitation measured by the Tropical Rainfall

671 Measuring Mission (TRMM) satellite instrument was adopted for wet deposition

672 screening, with a temporal resolution of 3 h and a spatial resolution of $0.25° \times 0.25°$.

673 We used the Hybrid Single Particle Lagrangian Integrated Trajectory (HYSPLIT,

674 version 4.9) model (http://www.ready.noaa.gov) to calculate the 48 h back trajectories

675 of the air masses arriving at NJU and PAES. The back trajectories were calculated

676 every 3 hour for July with the simulated layer heights of 50, 100 and 500 m above the

677 ground and the time step of 3 h (the same as the temporal resolution of TRMM). The

678 hourly accumulative precipitation along the 48 h back trajectories at two sites were

679 then calculated to determine the BC-CO data pairs influenced by precipitation, given

680 the little effect of precipitation on CO. Figure 11 illustrates the changes in the $\triangle$BC/

681 $\triangle$CO ratio observed at two sites for different accumulated precipitation intervals. At

682 NJU, the $\triangle$BC/$\triangle$CO ratio of air masses receiving less than 3 mm accumulated

683 precipitation was significantly larger than that of air masses receiving more than 3

684 mm, and the analogue number was 5 mm at PAES. In Case 7, therefore, we excluded

685 the BC-CO data pairs receiving more than 3 mm and 5 mm accumulated precipitation



along their trajectories within the last 48 h at NJU and PAES, respectively, in the
multiple regression model. It minimized the effect of wet deposition while retained
sufficient data points for the statistical significance. Figure 12 shows the simulated
wet deposition in Case 6 and the accumulated precipitation in Case 7 for July to
compare the data selection in the two cases. In Case 6, the number of data points were
reduced to 65% of Case 1 after data screening, and over 500 samples at the two sites
were available for the multiple regression model. In Case 7, only 31% of data points
remained. The periods excluded in Case 7 contained those in Case 6, implying a
stricter data screening to eliminate the effect of precipitation.
Table 9 shows the scaling factors estimated from the multiple regression model
in Case 7, and no big changes were found compared to the scaling factors for July in
Case 6 (Table 6). Consequently, the emissions by sector and total emissions in Case 7
were close to those in Case 6 (Table 7). The relative deviation of total emissions in
July between Case 7 and Case 1 (RD in Table 9) was 13%, and those for residential
and transportation were larger. The influence of precipitation was again indicated
insignificant, as the deviation was much smaller than that between the estimates
obtained from the bottom-up and top-down methods. Moreover, the CTM
performance based on Case 7, indicated by NMB and NME, was found similar to that
based on Case 6, implying the small effect of precipitation screening on simulation.
Even excluding the influence of precipitation along the back trajectories, the Sig. for
residential sources in Case 7 was still much larger than 0.05 (Table 9), suggesting
more efforts on quantification of emissions for this highly uncertain source category.
**5 Conclusions**
Monthly top-down estimates of BC emissions were derived from a multiple
regression model that integrated CTM and hourly BC concentrations from two ground
observation sites in southern Jiangsu city cluster. The annual emissions from
top-down approach (JS-posterior) were estimated at 13.4 Gg for 2015, 50.3% smaller
than those in bottom-up emission inventory that did not include the improved



emission controls in recent years (JS-prior), implying the effectiveness of air pollution
prevention measures on emission abatement. Application of JS-posterior in CTM
reduced the deviations between simulations and observations at two ground sites
effectively, especially at the urban site PAES. To evaluate the effects of observation
data on top-down estimate, two more cases in which observation data of only one site
(NJU) and observation data at both sites with their spatial representativeness
differentiated were applied to constrain the emissions, respectively. Best CTM
performance was found for the third case, indicating that inclusion of more ground
measurements with better spatiotemporal coverage in the city cluster would improve
the understanding of spatial distributions of BC emissions. In addition, top-down
estimates were derived from various bottom-up inventories, and the differences in
emission amount, spatial distribution and CTM performance between the constrained
emission estimates were significantly reduced compared to those between the
bottom-up inventories. The results implied that changes in initial emission input in the
regression model and CTM had limited effect on the top-down estimation. Finally, the
assumption of near-linearity between emissions and concentrations was justified, and
the influence of wet deposition on the estimated emissions was evaluated to be
moderate. This work demonstrated that top-down approach based on ground
observations and CTM could capture the fast changes in BC emissions attributed to
tightened pollution control policy at a city cluster scale. To further reduce uncertainty
of the approach, more ground measurements with sufficient temporal resolution
would be recommended at other regions in the city cluster. Data from other sources,
such as aerosol optical depth from satellite observation, could also be included to
improve the spatial and temporal distributions of emission estimates.

## Acknowledgement

This work was sponsored by the National Key Research and Development
Program of China (2017YFC0210106 and 2016YFC0201507), Natural Science




Foundation of China (91644220 and 41575142). We would like to acknowledge Tong

Dan from Tsinghua University for national emission data (MEIC).

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



**Figure captions**
Figure 1. Modeling domain and locations of two observation sites and three
meteorological stations.
Figure 2. The monthly (left axis) and annual emissions (right axis) by sector for
southern Jiangsu 2015 in JS-prior and JS-posterior (unit: Gg).
Figure 3. The seasonal variation of BC emissions by source (a) and total emissions (b)
in JS-prior, JS-posterior and MEIC-prior.
Figure 4. The observed and simulated hourly BC concentrations at NJU using JS-prior
and JS-posterior for January (a), April (b), July (c) and October (d) in 2015.
Figure 5. The same as Figure 4 but at PAES.
Figure 6. BC emission estimates by source of JS-prior, MEIC-prior, JS-posterior, and
MEIC-posterior in April 2015 in southern Jiangsu.
Figure 7. The spatial distributions of the deviations (JS-MEIC) between JS-prior and
MEIC-prior (a) and those between JS-posterior and MEIC-posterior (b).
Figure 8. The scatter plots of the simulated BC concentrations using JS inventories
versus those using MEIC at NJU (a) and PAES (b).
Figure 9. The correlation between the simulated BC concentrations with JS-prior and
those with doubled (a and c) or halved emissions in JS-prior (b and d) in July (a and b)
and October (c and d) at NJU. $F_{emis.}$ and $F_{conc.}$ indicate respectively the fraction of
changed emissions and that of changed simulated monthly average concentrations
between sensitivity and base simulation (Scenario B). $\triangle E$ and $\triangle C$ indicated the
change in emissions and that in simulated monthly average concentrations,
respectively.
Figure 10. The same as Figure 9 but at PAES.



Figure 11. The $\triangle$BC/$\triangle$CO ratio at NJU (a) and PAES (b) separated by different
accumulated precipitation along the back trajectories during 48 h. The number of
remaining data points is also given.
Figure 12. The wet deposition in Case 6 and accumulated precipitation in Case 7 at
NJU (a) and PAES (b). The number of remaining data points is also given.





## Tables

**Table 1. The scaling factors and statistical indicators from the multiple regression model for estimation of JS-posterior.**

| Month | Sector | Scaling factor | $t^a$ | Sig.$^b$ | VIF$^c$ | Sig.$^d$ |
|-------|--------|----------------|-------|----------|---------|----------|
| | Industry ($\beta_2$) | 0.42 | 2.65 | 0.01 | 1.76 | |
| January | Residential ($\beta_3$) | 1.31 | 3.67 | 0.00 | 2.37 | 0.00 |
| | Transportation ($\beta_4$) | 0.79 | 2.23 | 0.03 | 2.72 | |
| | Industry ($\beta_2$) | 0.22 | 0.96 | 0.34 | 2.65 | |
| April | Residential ($\beta_3$) | 0.58 | 1.63 | 0.11 | 4.62 | 0.00 |
| | Transportation ($\beta_4$) | 0.67 | 2.21 | 0.03 | 4.19 | |
| | Industry ($\beta_2$) | 0.35 | 3.09 | 0.00 | 2.09 | |
| July | Residential ($\beta_3$) | 0.39 | 0.95 | 0.34 | 2.95 | 0.00 |
| | Transportation ($\beta_4$) | 0.55 | 2.20 | 0.03 | 3.46 | |
| | Industry ($\beta_2$) | 0.34 | 1.92 | 0.06 | 1.53 | |
| October | Residential ($\beta_3$) | 1.52 | 4.12 | 0.00 | 2.20 | 0.00 |
| | Transportation ($\beta_4$) | 0.74 | 2.80 | 0.01 | 2.65 | |

Note: The criteria for the statistical significance of the model: a: t>2, b: Sig.<0.05, and

c: VIF<10, d: the overall significance.

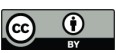



**Table 2. Statistical indicators for observed and simulated BC concentrations using JS-prior and JS-posterior at NJU and PAES.**

| Site | Parameter | January | | April | | July | | October | | Annual | |
|---|---|---|---|---|---|---|---|---|---|---|---|
| | | JS-prior | JS-posterior | JS-prior | JS-posterior | JS-prior | JS-posterior | JS-prior | JS-posterior | JS-prior | JS-posterior |
| NJU | Average SIM (µg/m³) | 5.97 | 5.50 | 2.38 | 1.82 | 1.99 | 1.29 | 2.80 | 2.42 | 3.44 | 2.82 |
| | Average OBS (µg/m³) | 5.44 | 5.44 | 2.69 | 2.69 | 2.65 | 2.65 | 3.96 | 3.96 | 3.83 | 3.83 |
| | NMB (%) | 8.35 | -0.08 | -16.02 | -32.40 | -23.09 | -51.32 | -29.20 | -39.01 | -10.16 | 26.43 |
| | NME (%) | 37.83 | 35.54 | 42.31 | 38.61 | 49.62 | 57.49 | 40.52 | 43.06 | 41.15 | 44.16 |
| | R | 0.67 | 0.66 | 0.34 | 0.43 | 0.36 | 0.31 | 0.42 | 0.48 | 0.67 | 0.69 |
| PAES | Average SIM (µg/m³) | 6.46 | 5.91 | 2.98 | 1.95 | 2.61 | 1.63 | 3.19 | 2.88 | 3.39 | 2.57 |
| | Average OBS (µg/m³) | 2.80 | 2.80 | 1.70 | 1.70 | 1.51 | 1.51 | 3.62 | 3.62 | 2.48 | 2.48 |
| | NMB (%) | 151.93 | 134.59 | 61.57 | 14.73 | 72.17 | 8.28 | -12.01 | -20.48 | 36.67 | 3.54 |
| | NME (%) | 155.53 | 139.50 | 73.18 | 42.87 | 92.74 | 42.37 | 43.10 | 40.80 | 72.00 | 57.55 |
| | R | 0.38 | 0.38 | 0.64 | 0.53 | 0.35 | 0.37 | 0.57 | 0.72 | 0.38 | 0.45 |

Note: SIM and OBS indicated the results from simulation and observation, respectively. NMB and NME were calculated using following
equations (P and O indicated the results from modeling prediction and observation, respectively):
$$NMB = \frac{\sum_{i=1}^{n}(P_i - O_i)}{\sum_{i=1}^{n}O_i} \times 100\%; \; NME = \frac{\sum_{i=1}^{n}|P_i - O_i|}{\sum_{i=1}^{n}O_i} \times 100\%$$



**Table 3. Statistical indicators for observed and simulated BC concentrations in**
**different cases in April 2015 at NJU and PAES.**

| Site | Parameter | Scenario B | Case1 | Case2 | Case3 | Case4 | Case5 |
|------|-----------|------------|-------|-------|-------|-------|-------|
| | Average SIM ($\mu g/m^3$) | 2.38 | 1.82 | 2.27 | 2.06 | 2.49 | 1.78 |
| | Average OBS ($\mu g/m^3$) | 2.69 | 2.69 | 2.69 | 2.69 | 2.69 | 2.69 |
| NJU | NMB (%) | -16.02 | -32.40 | -21.59 | -23.50 | -7.46 | -33.95 |
| | NME (%) | 42.31 | 38.61 | 32.47 | 32.64 | 41.58 | 38.94 |
| | R | 0.34 | 0.43 | 0.49 | 0.49 | 0.40 | 0.46 |
| | Average SIM ($\mu g/m^3$) | 2.98 | 1.95 | 2.45 | 2.01 | 5.13 | 2.29 |
| | Average OBS ($\mu g/m^3$) | 1.70 | 1.70 | 1.70 | 1.70 | 1.70 | 1.70 |
| PAES | NMB (%) | 61.57 | 14.73 | 49.86 | 18.02 | 201.35 | 34.71 |
| | NME (%) | 73.18 | 42.87 | 61.59 | 39.62 | 201.56 | 47.73 |
| | R | 0.64 | 0.53 | 0.63 | 0.66 | 0.65 | 0.59 |



**Table 4. The scaling factors and statistical indicators from the multiple**
**regression model in Case 3.**

| Site | Sector | Scaling factor | t | Sig. | VIF |
|------|--------|----------------|-----|------|-----|
| NJU | Industry ($\beta_2$) | 0.42 | 1.71 | 0.09 | 2.03 |
| | Residential ($\beta_3$) | 0.95 | 2.50 | 0.01 | 2.52 |
| | Transportation ($\beta_4$) | 0.65 | 2.13 | 0.03 | 2.66 |
| PAES | Industry ($\beta_2$) | 0.19 | 3.46 | 0.00 | 1.44 |
| | Residential ($\beta_3$) | 0.36 | 1.89 | 0.06 | 1.44 |
| | Transportation($\beta_4$) | 0.65 | - | - | - |



**Table 5. BC emissions from Nanjing and Suzhou–Wuxi–Changzhou-Zhenjiang**
**city cluster in different cases in April 2015 (Gg).**

| Case | Sector | Nanjing | Suzhou–Wuxi–Changzhou-Zhenjiang | Southern Jiangsu |
|---|---|---|---|---|
| Scenario B | Power | 0 | 0.01 | 0.01 |
| | Industry | 0.21 | 1.13 | 1.34 |
| | Residential | 0.08 | 0.24 | 0.32 |
| | Transportation | 0.12 | 0.30 | 0.42 |
| | Total | 0.41 | 1.68 | 2.09 |
| Case 1 | Power | 0 | 0.01 | 0.01 |
| | Industry | 0.05 | 0.25 | 0.30 |
| | Residential | 0.04 | 0.14 | 0.19 |
| | Transportation | 0.08 | 0.20 | 0.28 |
| | Total | 0.17 | 0.60 | 0.78 |
| Case 2 | Power | 0 | 0.01 | 0.01 |
| | Industry | 0.09 | 0.47 | 0.56 |
| | Residential | 0.07 | 0.23 | 0.30 |
| | Transportation | 0.08 | 0.20 | 0.27 |
| | Total | 0.24 | 0.91 | 1.14 |
| Case 3 | Power | 0 | 0.01 | 0.01 |
| | Industry | 0.04 | 0.47 | 0.51 |
| | Residential | 0.03 | 0.23 | 0.26 |
| | Transportation | 0.08 | 0.20 | 0.27 |
| | Total | 0.15 | 0.90 | 1.05 |



**Table 6. The scaling factors and statistical indicators from the multiple**
**regression model in Case 6.**

| Month | Sector | Scaling factor | $t^{a}$ | Sig.[b] | VIF[c] | Sig.[d] |
|---|---|---|---|---|---|---|
| | Industry ($\beta_2'$) | 0.41 | 2.17 | 0.03 | 1.71 | |
| January | Residential ($\beta_3'$) | 1.53 | 3.48 | 0.00 | 2.29 | 0.00 |
| | Transportation ($\beta_4'$) | 0.73 | 1.65 | 0.10 | 2.66 | |
| | Industry ($\beta_2'$) | 0.24 | 0.92 | 0.36 | 1.91 | |
| April | Residential ($\beta_3'$) | 0.51 | 1.32 | 0.19 | 3.29 | 0.00 |
| | Transportation ($\beta_4'$) | 0.70 | 2.12 | 0.03 | 3.03 | |
| | Industry ($\beta_2'$) | 0.38 | 4.43 | 0.00 | 1.43 | |
| July | Residential ($\beta_3'$) | 0.34 | 0.82 | 0.41 | 2.52 | 0.00 |
| | Transportation ($\beta_4'$) | 0.74 | 3.55 | 0.00 | 2.25 | |
| | Industry ($\beta_2'$) | 0.33 | 1.00 | 0.32 | 1.44 | |
| October | Residential ($\beta_3'$) | 1.36 | 2.61 | 0.01 | 1.86 | 0.00 |
| | Transportation ($\beta_4'$) | 0.72 | 1.89 | 0.06 | 2.02 | |

Note: The criteria for the statistical significance of the model: a: $t>2$, b: Sig.$<0.05$, and
c: VIF$<10$, d: the overall significance.





**Table 7. The monthly and annual emissions by sector for southern Jiangsu 2015 in Case 6 (unit: Gg) and the relative deviation compared to Case 1 (RD: Case 6–Case 1)/Case 1).**

| | January | | April | | July | | October | | Annual | |
|---|---|---|---|---|---|---|---|---|---|---|
| | Case 6 | RD | Case 6 | RD | Case 6 | RD | Case 6 | RD | Case 6 | RD |
| Power | 0.0 | 0.0% | 0.0 | 0.0% | 0.0 | 0.0% | 0.0 | 0.0% | 0.0 | 0.0% |
| Industry | 0.6 | -2.4% | 0.3 | 9.9% | 0.6 | 9.2% | 0.5 | -0.3% | 6.0 | 3.1% |
| Residential | 0.5 | 16.7% | 0.2 | -13.1% | 0.1 | -13.7% | 0.4 | -10.2% | 3.6 | -0.6% |
| Transportation | 0.3 | -8.2% | 0.3 | 4.3% | 0.4 | 34.4% | 0.3 | -3.0% | 3.9 | 5.4% |
| Sum | 1.4 | 2.4% | 0.8 | 2.3% | 1.1 | 13.6% | 1.2 | -4.2% | 13.5 | 2.6% |







**Table 8. Statistical indicators for the observed and simulated BC concentrations**
**in July 2015 at NJU and PAES in Case 6 and Case 7.**

| | Parameter | Case 6 | Case 7 |
|---|---|---|---|
| NJU | Average SIM ($\mu g/m^3$) | 1.40 | 1.41 |
| | Average OBS ($\mu g/m^3$) | 2.65 | 2.65 |
| | NMB (%) | -47.41 | -46.72 |
| | NME (%) | 54.88 | 54.44 |
| | R | 0.33 | 0.33 |
| PAES | Average SIM ($\mu g/m^3$) | 1.76 | 1.76 |
| | Average OBS ($\mu g/m^3$) | 1.51 | 1.51 |
| | NMB (%) | 16.87 | 16.65 |
| | NME (%) | 44.46 | 42.71 |
| | R | 0.36 | 0.39 |

Note: SIM and OBS indicated the results from simulation and observation,
respectively. NMB and NME were calculated using following equations (P and O
indicated the results from modeling prediction and observation, respectively):
$$NMB = \frac{\sum_{i=1}^{n}(P_i - O_i)}{\sum_{i=1}^{n}O_i} \times 100\% \qquad NME = \frac{\sum_{i=1}^{n}|P_i - O_i|}{\sum_{i=1}^{n}O_i} \times 100\%$$
;


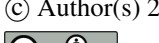



**Table 9. The scaling factors and statistical indicators from the multiple regression model in Case 7. The emissions by sector for southern Jiangsu 2015 July in Case 7 (unit: Gg) and the relative deviations (RD) compared to Case 1 (RD: Case 7–Case 1)/Case 1) are also shown in table.**

| Sector | Scaling factor | $t^a$ | Sig.$^b$ | VIF$^c$ | Sig.$^d$ | Emissions | RD |
|---|---|---|---|---|---|---|---|
| Power | | | | | | 0.0 | 0.0% |
| Industry ($\beta_2'$) | 0.38 | 2.38 | 0.02 | 1.31 | | 0.5 | 9.5% |
| Residential ($\beta_3'$) | 0.31 | 0.31 | 0.75 | 2.31 | 0.00 | 0.1 | -20.6% |
| Transportation ($\beta_4'$) | 0.75 | 1.8 | 0.07 | 1.95 | | 0.4 | 36.4% |
| Sum | | | | | | 1.0 | 13.4% |

Note: The criteria for the statistical significance of the model: a: t>2, b: Sig.<0.05, and

c: VIF<10, d: the overall significance.



1037    **Figure 1**

1038

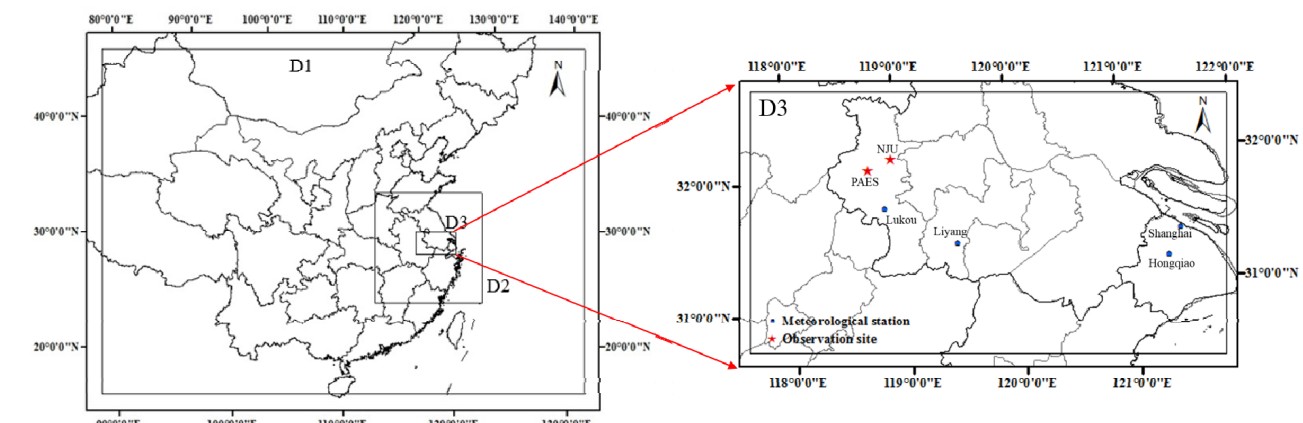

1039



**Figure 2**

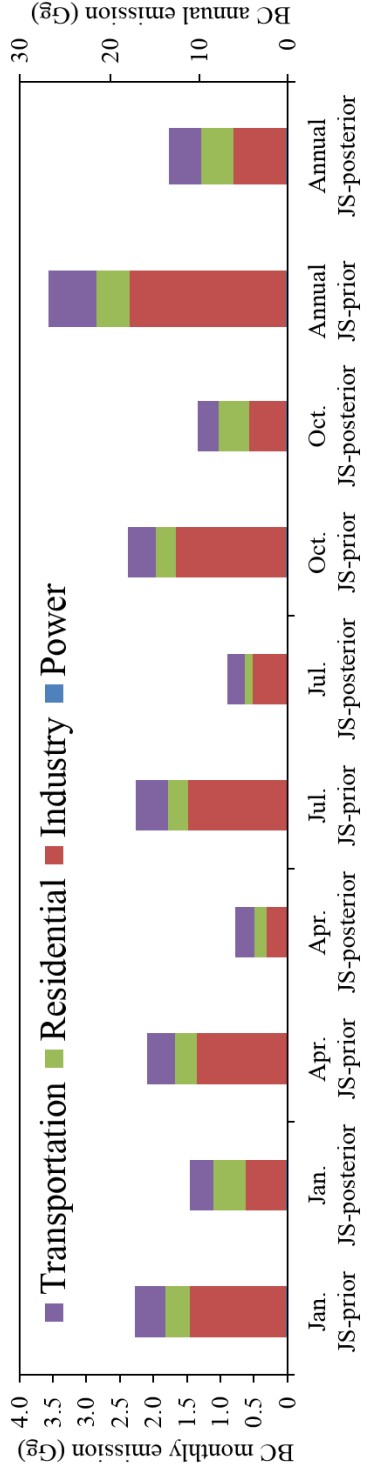





**Figure 3**

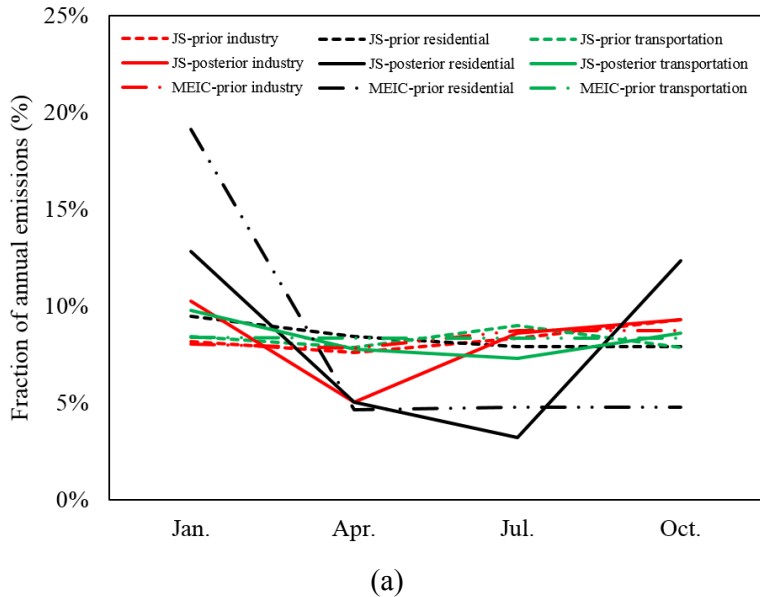

(a)

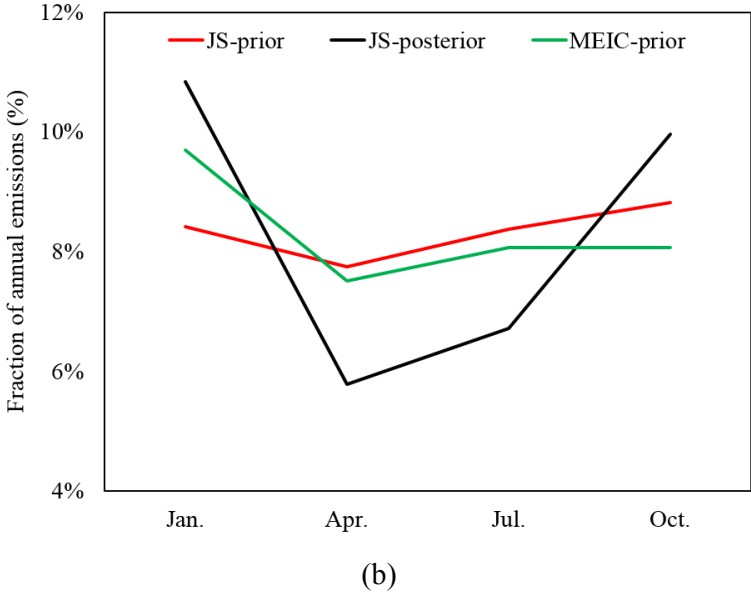

(b)

**Figure 4**

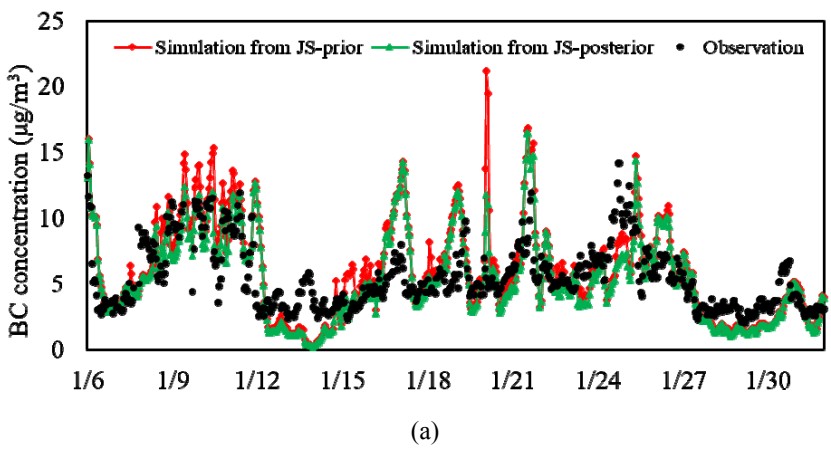

(a)

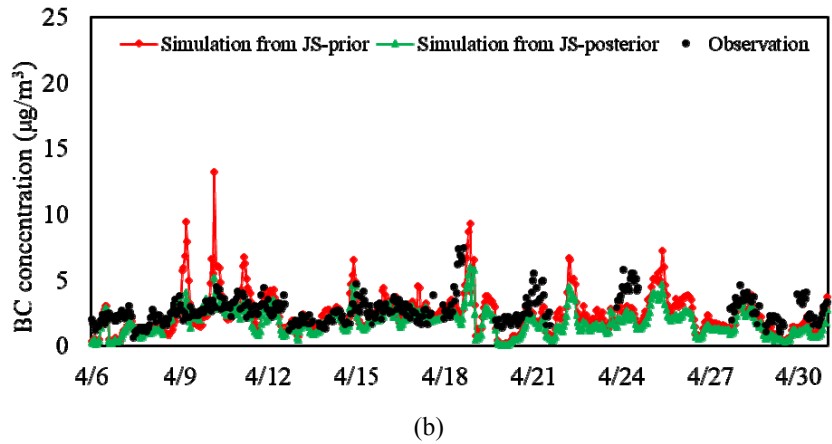

(b)

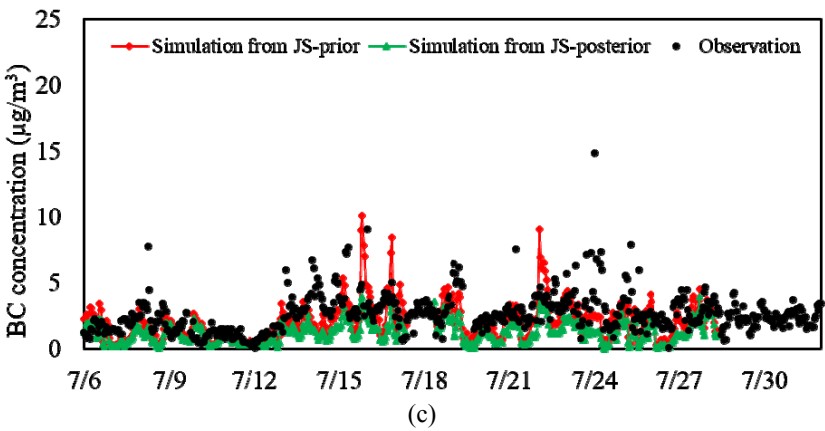

(c)





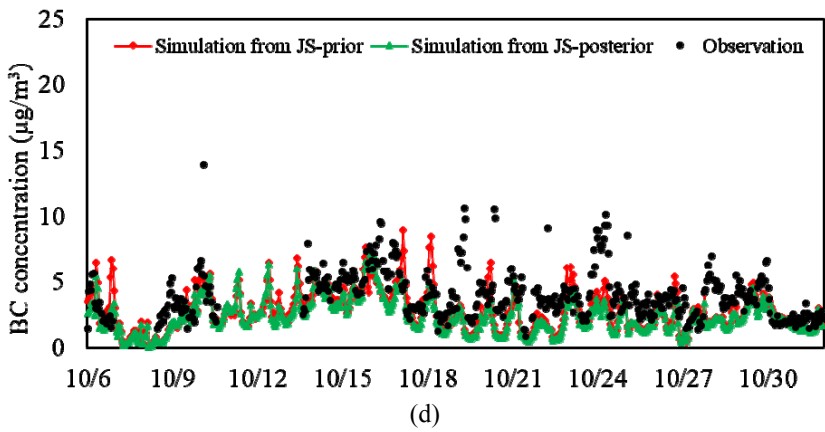

(d)

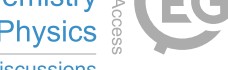

**Figure 5**

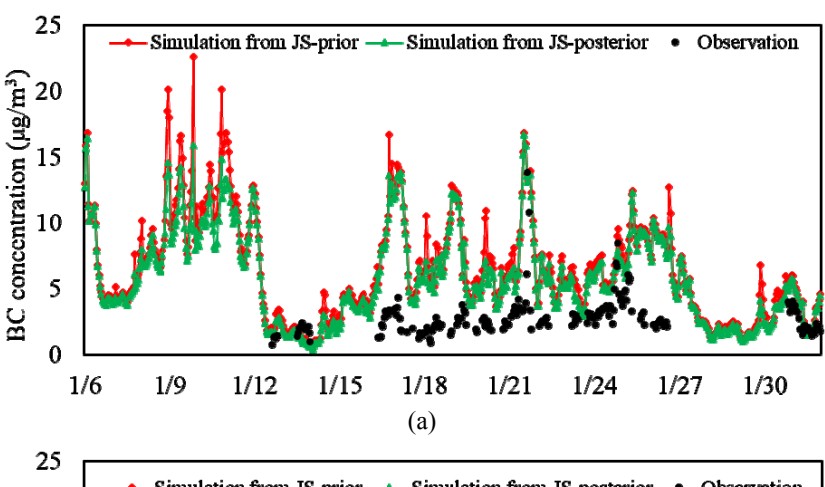

(a)

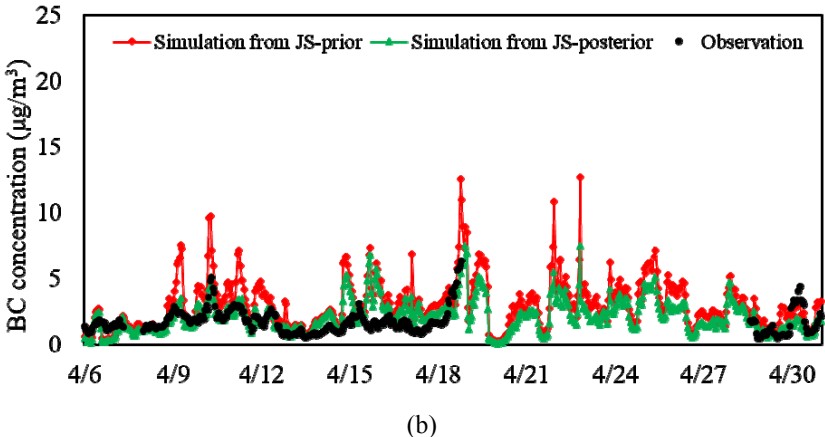

(b)

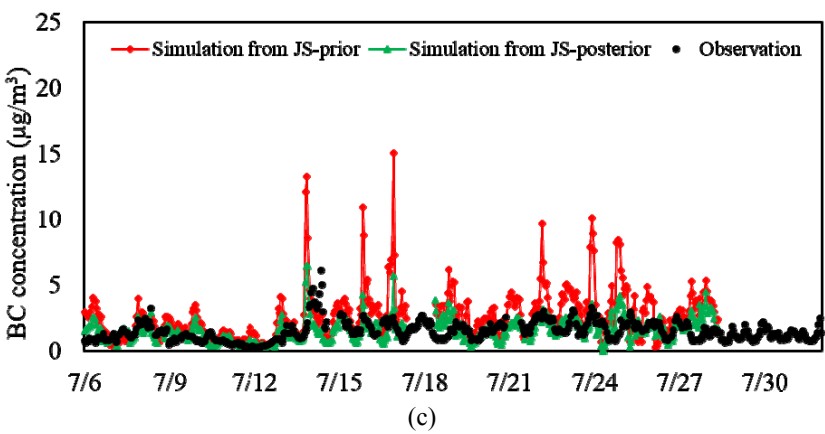

(c)





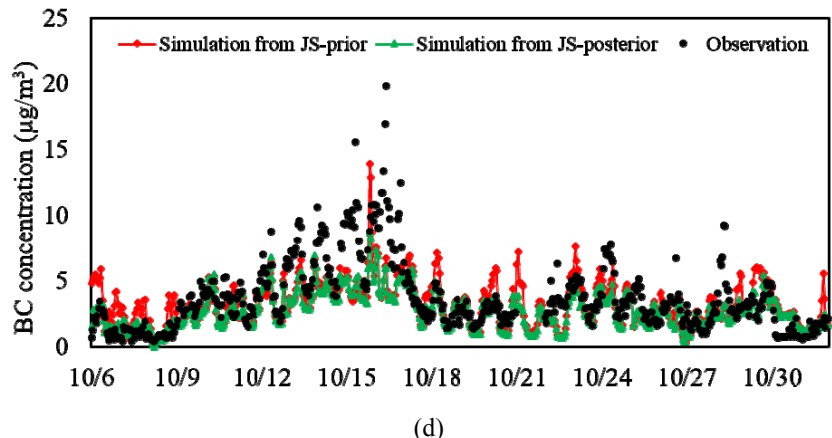

(d)




**Figure 6**

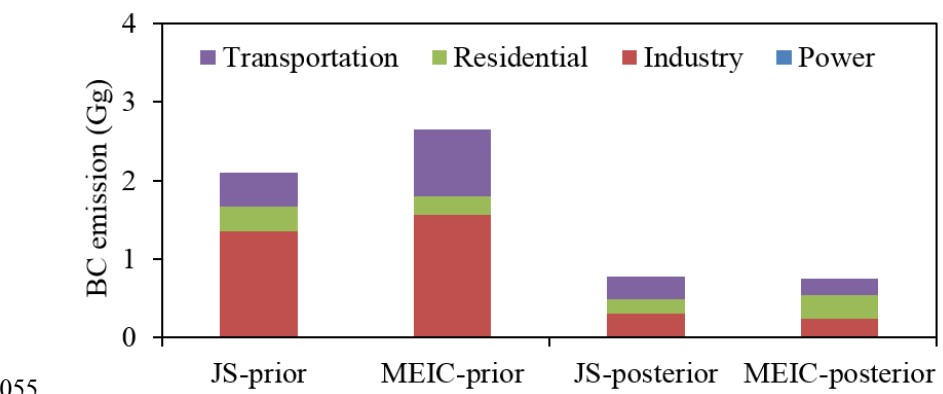




**Figure 7**

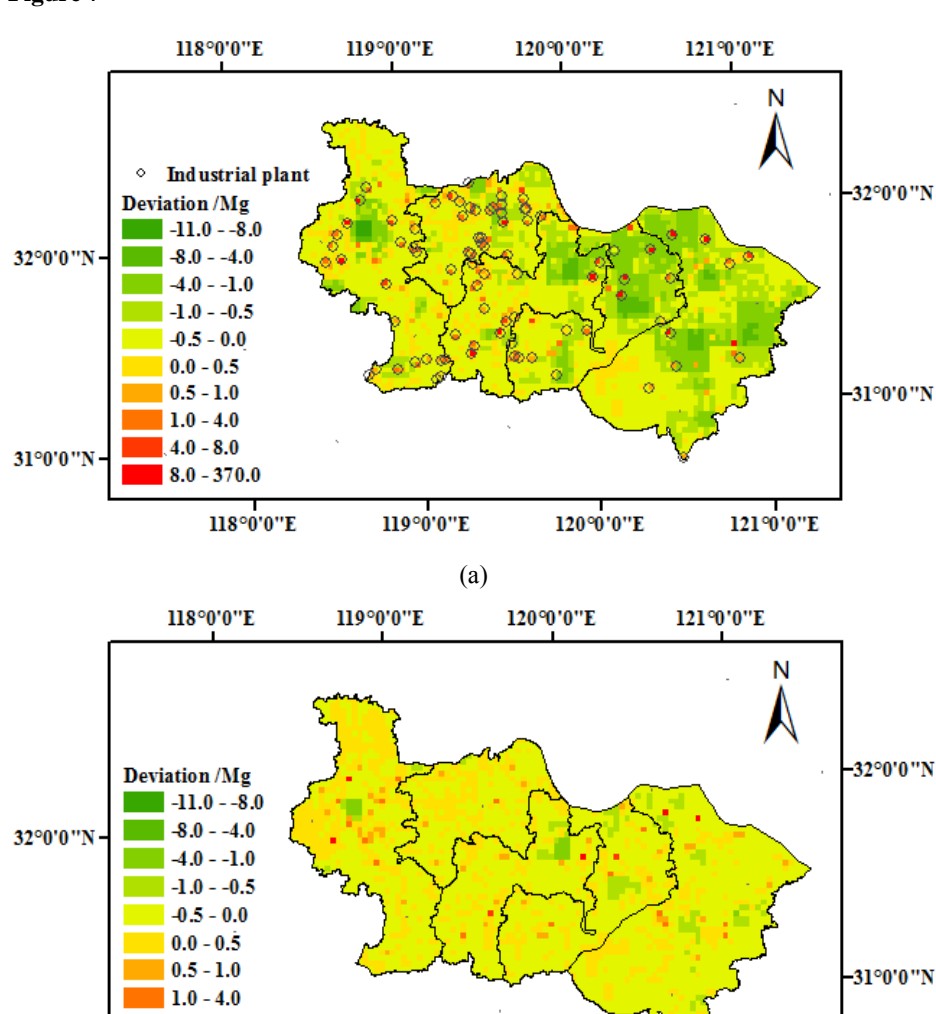

(a)

(b)






**Figure 8**

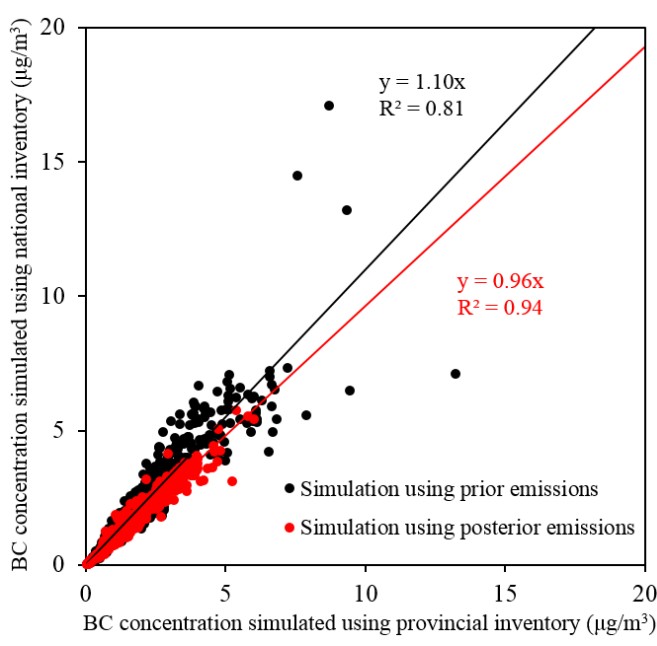

(a)

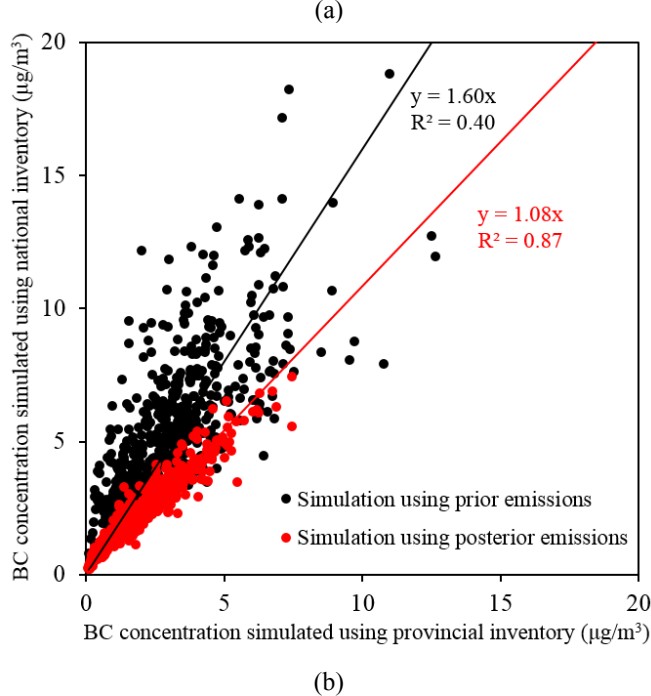

(b)





**Figure 9**

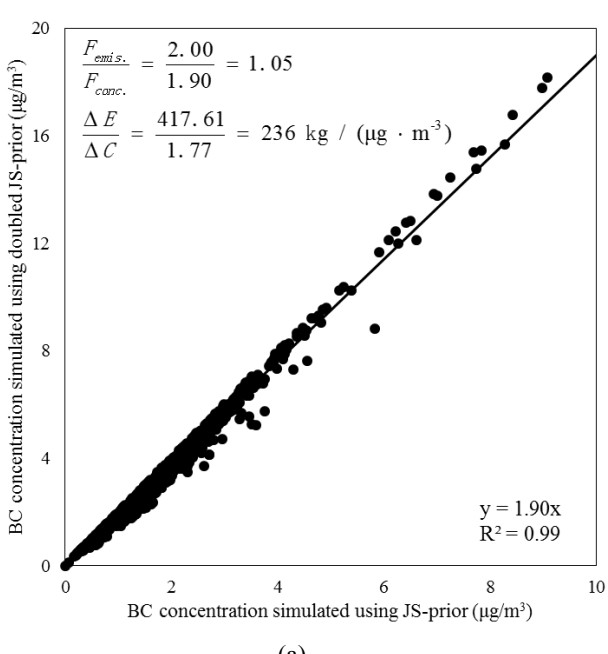

(a)

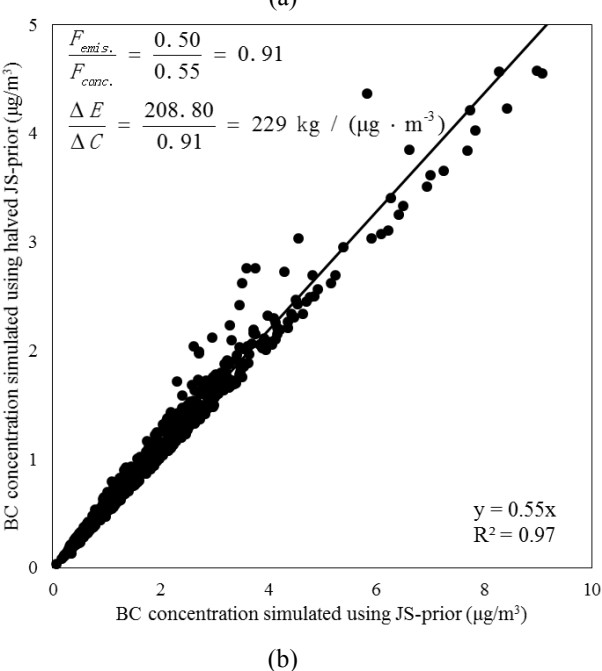

(b)


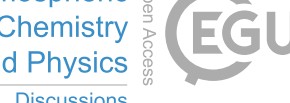

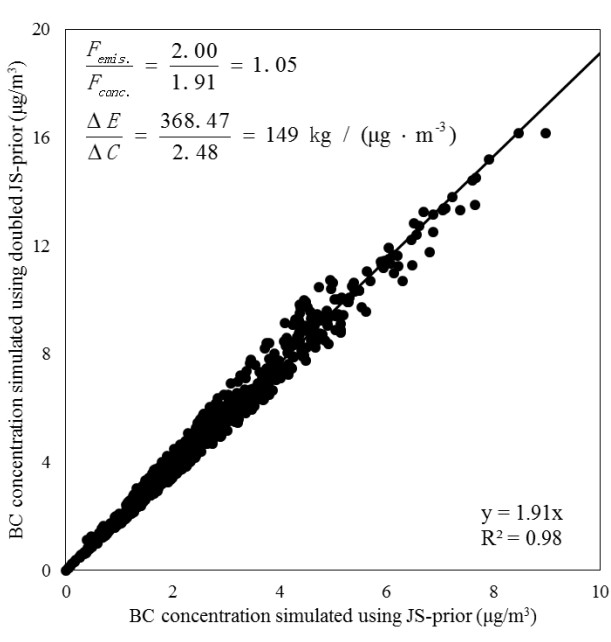

(c)

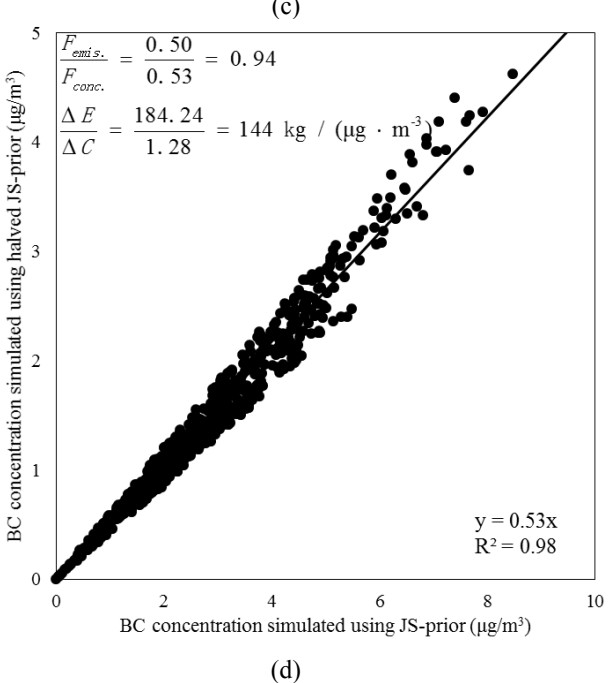

(d)




**Figure 10**

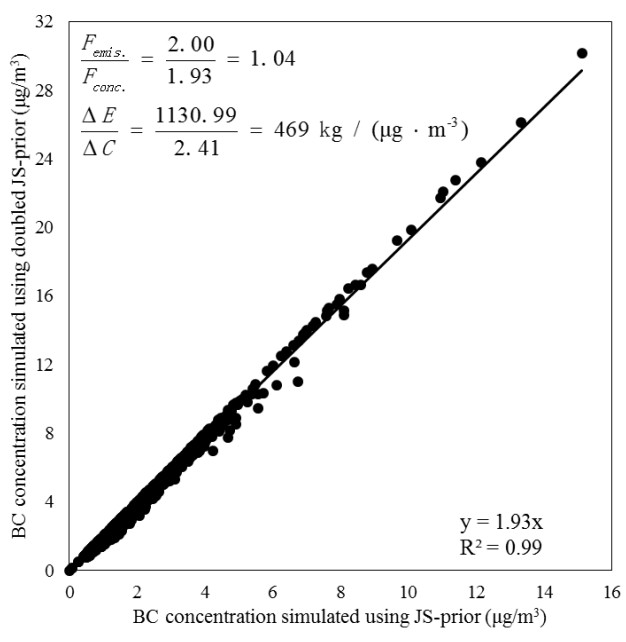

(a)

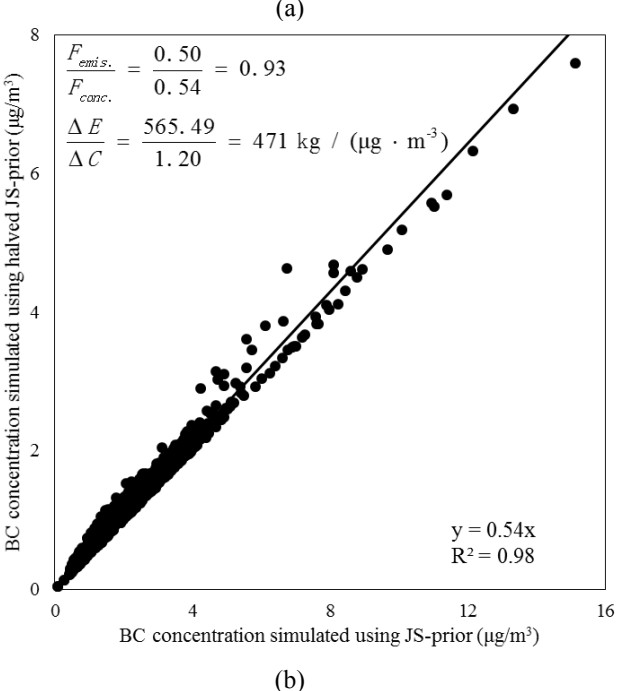

(b)






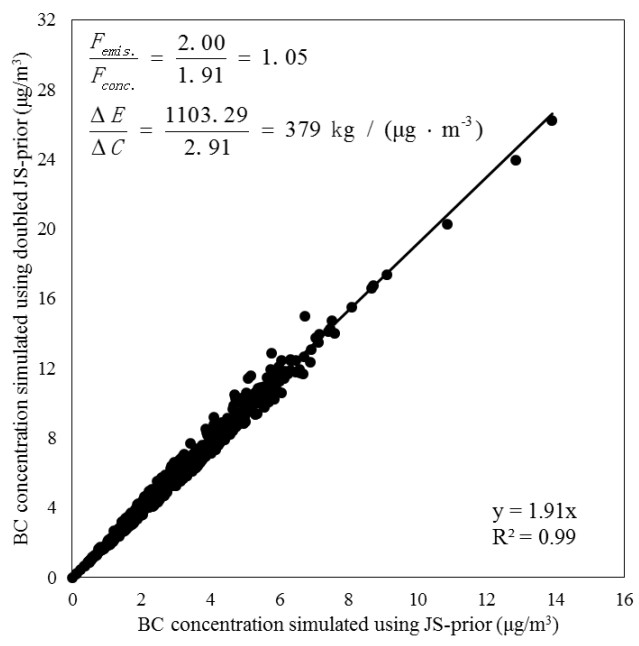

(c)

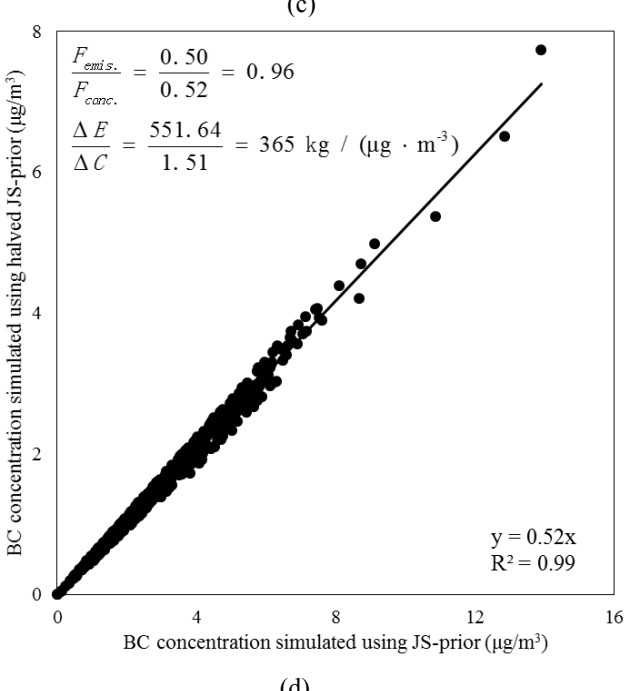

(d)







**Figure 11**

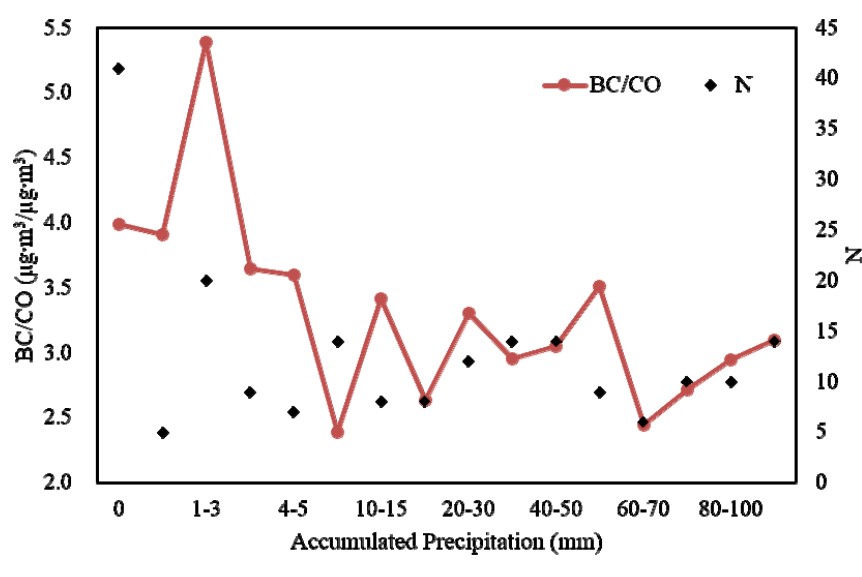

(a)

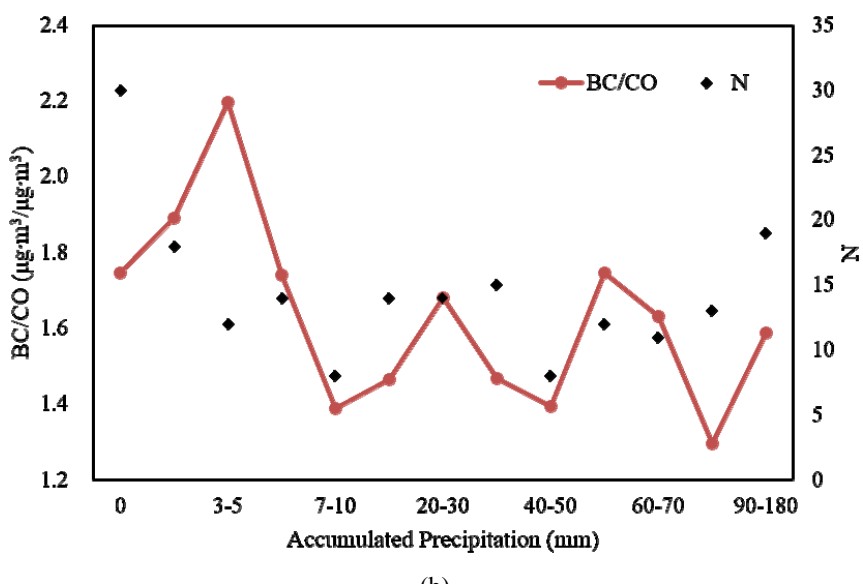

(b)




**Figure 12**

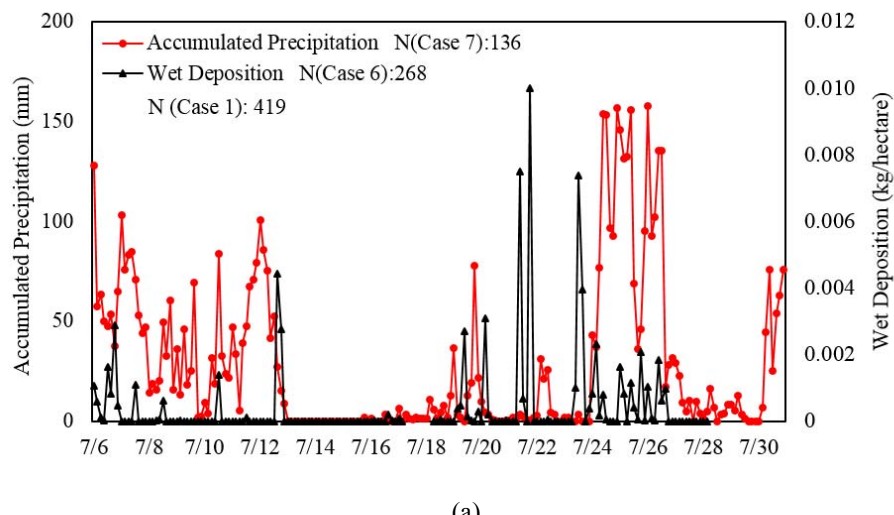

(a)

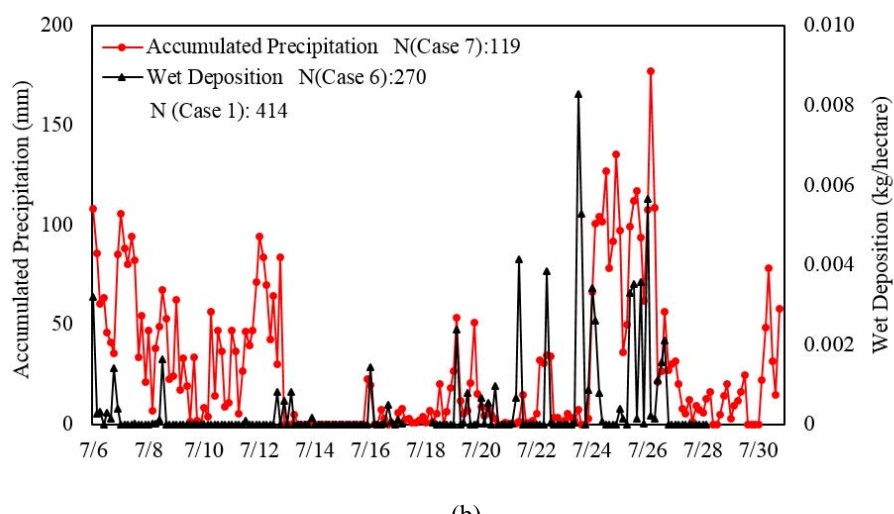

(b)
