# Peer review of "Top-down estimate of black carbon emissions for city cluster"

_Atmospheric Chemistry and Physics, 2018_

## Referee Comment (RC1) · Anonymous Referee #1 · 21 Oct 2018

General Comments:

The authors provide a detailed analysis to constrain BC emissions from Jiangsu (China) using observations from two stations. They found BC emissions are significantly overestimated in the bottom-up inventories, which has important implications. However, I have some major concerns about the representation of their stations to the whole region, and the inversion methodology. I recommend the paper for publication after consideration of the points below.

Specific Comments:

1: Abstract Lines 28-29, please confirm the same BC concentrations (i.e. 3.4 ug/m3)

[Figure]

at both sites. In addition, Lines 39-40 say: "the simulated annual mean was elevated to 2.6". I assume it is elevated from 3.4 to 2.6?

2: Line 257-258 Are 5 days long enough to minimize the influences of initial conditions? I checked the methodology of other studies and found much longer initialization periods. For example, 3 months in Wang et al. (2013) and Mao et al. (2015).

Wang, X., Wang, Y., Hao, J., Kondo, Y., Irwin, M., Munger, J. and Zhao, Y.: Top‐-down estimate of China's black carbon emissions using surface observations: Sensitivity to observation representativeness and transport model error, J Geophys Res Atmospheres, 118(11), 5781–5795, doi:10.1002/jgrd.50397, 2013.

Mao, Y., Li, Q., Chen, D., Zhang, L., Hao, W. and Liou, K.: Top-down estimates of biomass burning emissions of black carbon in the Western United States, Atmos Chem Phys, 14(14), 7195–7211, doi:10.5194/acp-14-7195-2014, 2014.

3: Table 2 As shown with the annual mean result:

* NJU, the a priori is 0.4 lower than obs, and is reduced by 0.6 in the inversion. The a posteriori is 1.0 lower than obs.

* PAES, the a priori is 0.9 higher than obs, and is reduced by 0.8 in the inversion. The a posteriori is 0.1 higher than obs.

It seems that the inversion simply moves the bias from PAES to NJU by reducing the total emissions, suggesting the inversion system is dominated by PAES. Considering the inconsistency between NJU and PAES, it is hard to say whether the conclusion is reliable to provide a good representation for the whole region.

4: Section 4.1 As shown in Table 2, the model simulation (2.38) is already lower than obs (2.69) in April at NJU. When only NJU data is used, how could the inversion keep reducing the emissions with scaling factors, 0.42, 0.95 and 0.65? Theoretically, an inversion system should minimize the discrepancy between model and obs rather than magnifying it.

5: Table 4 More information is needed in the caption. It is really difficult to follow the discussion to distinguish the Cases (B, 1, 2, 3, 4, 5) and Cases (6, 7).

---

## Referee Comment (RC2) · Anonymous Referee #2 · 12 Nov 2018

In this study, Zhao et al. uses ground-based elemental carbon (EC) measurements from two sites in eastern China to evaluate and constrain black carbon (BC) emissions from two bottom-up inventories: a national/regional inventory for China (MEIC) and a high-resolution inventory for city clusters in southern Jiangsu Province. Both inventories include emissions from transportation, industry, power generation, and the residential sector. The authors show that the posterior emission estimates, constrained by ground measurements, are much smaller than the prior emission estimates, suggesting that pollution control measures by the Jiangsu government have effectively reduced emissions of BC. They also show results from various sensitivity tests, including those on the number of observation sites, spatial representativeness of observation sites, a

priori emission inventories, and wet deposition. Overall, this is an interesting study that can be potentially useful for air quality modeling and management, emission inventory development and evaluation, and also studies on regional aerosol effects. Through several fairly detailed sensitivity tests, the authors also demonstrate that the differences between a priori and posteriori emission estimates are robust. However, the paper is overly long (and needs some improvement in presentation quality) and some reorganization may help. And there are also some concerns about the methodology that need to be addressed before this paper can be published in ACP.

Major comments: It is not quite clear whether emissions outside of Jiangsu Province (but within the model domains) are scaled or not. Given the location of the sites, they could be strongly influenced by emissions from nearby provinces. If different local governments implemented different pollution control measures but the same domain wide scaling factors are used for emissions, that may lead to biases in the final estimated emissions for southern Jiangsu.

The lack of biomass burning emissions can be concerning. Could the model underestimates of BC in July and particularly October be caused by the biomass burning (particularly agricultural fires)? How does the lack of biomass burning emissions affect the estimated emissions for other sectors?

The paper is overly long and can be better organized. In particular, if spatial representativeness and wet deposition are important, can the authors focus on the top-down estimates that consider both of these factors? Description of the other sensitivity tests can be brief. Also writing needs to be improved.

Specific comments: Figure 3 and the paragraph starting from line 389: given that the scaling factor for April and Oct. are more uncertain (in terms of their statistical significance), are the seasonal patterns in the posterior emission estimates significant?

Figure 5a – what may have caused the model overestimates in mid-January at PAES? How does this period affect emission estimates? Can the authors exclude this period

and compare the top-down estimates?

Lines 456-461: again, could the model bias be due to the lack of biomass burning emissions?

Tables: There are already many tables in the paper (and maybe not everyone is absolutely necessary). But a table that summarizes the different cases may be helpful for readers to keep track.

Table 4 and related discussion on case 3: would the authors expect somewhat different driving conditions and emission factors for automobiles in urban and suburban settings? If so, is it still a valid assumption to assume the same scaling factor between NJU and PAES for transportation?

---

## Referee Comment (RC3) · Anonymous Referee #3 · 19 Nov 2018

This is a very nice and detailed work to constrain BC emissions in southern Jiangsu. The approach and uncertainty analysis may be applied to other regions. The paper is well written in general, suitable for ACP. Below are a few suggestions to further improve the paper.

It would be nice to discuss in the conclusion section the potential of applying the method to other regions.

The regression model needs to be further clarified. Are the scaling factors (betta) for each month, day, or hour? Why is there not a term in Eq. 1 for the background (e.g., lateral boundary condition) reflecting the effect of horizontal transport from regions

other than southern Jiangsu? Table S3 and Fig. S7 show that the sum of southern Jiangsu contributions is much smaller than 100%, implying a large contribution from regions other than southern Jiangsu.

The idea of testing the spatial representativeness of measurements is very nice. Given the spatial representativeness difference between the two sites, is it possible to use Case 3 as your best case? Alternatively, it would be nice to improve the regression model by taking into account the transport path, e.g., by basing on WRF modeled winds to design a model that considers the trajectory of air movement. The much higher bias in JS-posterior than JS-prior in Case 1, which is a concern, is related to this spatial representativeness issue.

A clearer discussion of temporal resolution in bottom up inventories and how this resolution affects the top-down constraint will be very helpful.

Comparison with near-surface measurements is sensitive to WRF/CMAQ modeled vertical processes, including the number of vertical layers within the PBL, the thickness of the first layer, and the model error in vertical mixing representation. WRF/CMAQ may have some issues with PBL mixing (Liu et al., 2018). Please specify these model setups. Please discuss the potential effect of model vertical resolution/mixing/transport errors on the BC constraint.

Table S2 shows that the prevailing winds in all three meteorological sites are southerly or southeasterly. I thought there would be northerly in the cold months (January and October). Please double check.

Minor comments:

Some paragraphs are too long and should be splitted, for example, L71-111, L352-388.

Abstract – please specify that monthly, sector-level and city-level emissions are optimized.

L22 – "observations," should be "observations" (no comma)

Abstract – please specify that WRF/CMAQ is used

L214 – is there is term for background (due to horizontal transport)?

L218 – "domain-wide" – here you optimize the southern Jiangsu emissions, not the domain-wide emissions. Also, as suggested above, an improved regression model may be used to better account for spatial representativeness of measurements.

L256 – "coordinated" should be "coordinate"

L274-288 – please specify the temporal resolution of bottom up emissions.

L283-285 – do you remove emissions in the whole domain, or just southern Jiangsu cities?

L288 – "Scenarios B and S" should be "Scenarios B and S1-S4"

L324 – "double" should be "twice"

L340 – "VIF smaller than 10" – the VIF values in the table are much smaller than 10.

L386-388 – this sentence is not clear

L418-442 – A figure would be much better than a table for this type of analysis.

L426 – what do you mean by "commonly"? The wording may be improved.

L443-446 – The increased bias from JS-prior to JS-posterior at NJU should be discussed in more detail.

L464 – some cases are for other months.

L551 – "initial" should be "a priori". Please revise throughout the text.

L573-604 – the paragraph contains multiple messages, and is better to be splitted.

Figs. S8-11 – the dates of precipitation are also not very well simulated.

L701 – "insignificant" should be "modest"

L715-717 – the increased bias at NJU should be mentioned.

L735-737 – it would be extremely difficult to use satellite AOD to constrain BC emissions.

References: Liu et al., Spatiotemporal variability of NO2 and PM2.5 over Eastern China: observational and model analyses with a novel statistical method, Atmospheric Chemistry and Physics, 18, 12933-12952, doi:10.5194/acp-18-12933-2018, 2018.
* * *

---

## Author Response (AR1)

**Main revisions and response to reviewers' comments**

Manuscript No.: acp-2018-984

Title: Top-down estimate of black carbon emissions for city cluster using ground observations: A case study in southern Jiangsu, China

Authors: Xuefen Zhao, Yu Zhao, Dong Chen, Chunyan Li, Jie Zhang

We thank very much for the valuable comments and suggestions from the three reviewers, which help us improve our manuscript significantly. The comments were carefully considered and revisions have been made in response to suggestions. Following is our point-by-point responses to the comments and corresponding revisions.

**Reviewer #1**

*0. The authors provide a detailed analysis to constrain BC emissions from Jiangsu (China) using observations from two stations. They found BC emissions are significantly overestimated in the bottom-up inventories, which has important implications. However, I have some major concerns about the representation of their stations to the whole region, and the inversion methodology. I recommend the paper for publication after consideration of the points below.*

**Response and revisions:**

We appreciate the reviewer's remarks on the importance of the work. Regarding the limitations pointed out by the reviewer, we have improved the manuscript accordingly. The spatial representativeness of the two sites in the multiple regression model has been clearly described (please see our response to Q3). Case 2 in which observation data at only one site (NJU) were used has been further re-analysed to avoid confusion to the inversion methodology (please see our response to Q4).

***1.*** *Abstract Lines 28-29, please confirm the same BC concentrations (i.e. 3.4 ug/m³) at both sites. In addition, Lines 39-40 say: "the simulated annual mean was elevated to 2.6". I assume it is elevated from 3.4 to 2.6?*

**Response and revisions:**

We thank the reviewer's comment and reminder. We confirmed that the annual mean simulations of BC were 3.44 and 3.39 ug/m³ at NJU and PAES, respectively. When the constrained emissions were applied, the annual mean concentration was simulated to decrease from 3.39 to 2.57 ug/m³ at PAES, and it was indicated **in Table 2 in the revised manuscript**. We corrected the sentence **in line 41 in the revised manuscript**: "At PAES, in particular, the simulated annual mean declined to 2.6 μg/m³ and the annual normalized mean error (NME) decreased from 72.0% to 57.6%."

***2.*** *Line 257-258 Are 5 days long enough to minimize the influences of initial conditions? I checked the methodology of other studies and found much longer initialization periods. For example, 3 months in Wang et al. (2013) and Mao et al. (2015).*

**Response and revisions:**

We thank and agree with the reviewer's comment. Some studies that applied GEOS-Chem or WRF-Chem to constrain BC emissions at larger spatial scale often chose several months as spin up to minimize the influence of initial conditions (Fu et al., 2013 and studies mentioned by the reviewer). For WRF-CMAQ model, in contrast, more studies used several days as initialization periods, for example, 5 days in Chang et al. (2018) and Tran et al. (2018), and 7 days in Ran et al. (2016). The period in this study is expected to be sufficient to minimize the influence of initial condition.

*3.* *Table 2 As shown with the annual mean result:*

*\* NJU, the a priori is 0.4 lower than obs, and is reduced by 0.6 in the inversion. The a posteriori is 1.0 lower than obs.*

*\* PAES, the a priori is 0.9 higher than obs, and is reduced by 0.8 in the inversion. The a posteriori is 0.1 higher than obs.*

*It seems that the inversion simply moves the bias from PAES to NJU by reducing the total emissions, suggesting the inversion system is dominated by PAES. Considering the inconsistency between NJU and PAES, it is hard to say whether the conclusion is reliable to provide a good representation for the whole region.*

**Response and revisions:**

We appreciate the reviewer's important comment. As can been seen **in Table 2 and Figures 3 and 4 in the revised manuscript**, application of JS-posterior effectively reduced the large biases between simulations and observations for all seasons at PAES and for January and April at NJU, suggested by the reduced NMEs. In particular, most of the overestimations in peak concentrations were corrected at the both sites. We mentioned **in lines 489-492, 497-499 and 508 in the revised manuscript**. It should be also acknowledged that NMEs for July and October and the annual average of NME were slightly enhanced at NJU. Limitation of the multiple regression model was thus indicated that overestimation and underestimation in concentrations at different sites could hardly be corrected simultaneously without further improvement in spatial distribution of emissions, and we mentioned in details **in lines 511-516 in the revised manuscript**.

To improve the method and to quantify the effect of spatial representation of observation sites on top-down estimate, we provided Case 3 in which observation data at PAES and NJU were applied to constrain emissions from Nanjing and Suzhou-Wuxi-Changzhou-Zhenjiang city cluster, respectively, **in Section 4.1 in the revised manuscript**. The best CTM performance was obtained in Case 3, implying that inclusion of more measurement data with their spatial representativeness considered could improve the top-down method. Given the limited BC observation data in the area, therefore, more measurements with better spatiotemporal coverage were recommended for constraining BC emissions effectively, as mentioned **in lines 47-52 in the revised manuscript**.

**4.** *Section 4.1 As shown in Table 2, the model simulation (2.38) is already lower than obs (2.69) in April at NJU. When only NJU data is used, how could the inversion keep reducing the emissions with scaling factors, 0.42, 0.95 and 0.65? Theoretically, an inversion system should minimize the discrepancy between model and obs rather than magnifying it.*

**Response and revisions:**

We thank the reviewer's important comment. As can been seen **in Table 2 in the revised manuscript**, the monthly mean of simulated BC concentrations at NJU with JS-prior was 2.38 ug/m$^3$ for all periods in April, smaller than the observed 2.69 ug/m$^3$. For Case 2 in which only NJU data were applied, the scaling factors for industry, residential and transportation emissions were obtained at 0.42, 0.95, and 0.65, respectively, implying a further reduction in BC emissions. The main reason is that the data for the whole April were not fully used due to necessary data screening in the multiple regression model. We acknowledge that the data screening process was not clearly stated in the original manuscript. Before applying in the multiple regression model, we excluded the periods following the criterions: the periods lack of observation data, those for which the contribution of each emission sector (power generation, industry, residential sources and transportation) was simulated to be smaller than zero through the brute-force method, and those for which the sum of contributions of all the four sectors was larger than 100% with CTM. The data screening helped to reduce the uncertainty of CTM in the multiple regression model. We added the description of data screening **in lines 240-245 in the revised**

**manuscript**. The number of data after screening in Case 2 was 48% of data in all periods (most data screening was due to lack of observations, accounting for 38%). We divided all the data points in April in Case 2 into two groups: those included in the multiple regression model and those excluded from the model, and analyzed the modeling performances for both groups separately. As can be seen in Table R1, the simulated concentration for periods included in the multiple regression model (2.71 $\mu g/m^3$) was larger than the observation (2.56 $\mu g/m^3$) when JS-prior was applied, different from the case without data screening (i.e., data in all periods were included). The emissions could then be reduced when the observation was applied in the constraining. As a result, application of the top-down estimate in Case 2 effectively reduced the NME for the period included in the model from 34.01% to 21.09%, and the simulated average concentration was closer to the observation. At the same time, the constrained emissions did not increase the bias for periods excluded from the multiple regression model. It thus indicated that the underestimation for periods excluded from the multiple regression model could result largely from factors other than emissions like meteorology. We added the analysis **in lines 547-559 in the revised manuscript** and included Table R1 **as Table S8 in the revised supplement**.

Table R1. Statistical indicators for observed and simulated BC concentrations for all periods, those included in the multiple regression model, and those excluded from the model in JS-prior and Case 2 for April 2015 at NJU.

| Site | Parameter | JS-prior: All period | JS-prior: Included | JS-prior: Excluded | Case 2: All period | Case 2: Included | Case 2: Excluded |
|------|-----------|------|------|------|------|------|------|
| NJU | Average SIM ($\mu g/m^3$) | 2.38 | 2.71 | 2.08 | 2.27 | 2.42 | 2.08 |
|  | Average OBS ($\mu g/m^3$) | 2.69 | 2.56 | 2.99 | 2.69 | 2.56 | 2.99 |
|  | NMB (%) | -16.02 | 5.90 | -56.48 | -21.59 | -5.32 | -56.63 |
|  | NME (%) | 42.31 | 34.01 | 57.62 | 32.47 | 21.09 | 57.61 |

**5.** *Table 4 More information is needed in the caption. It is really difficult to follow the discussion to distinguish the Cases (B, 1, 2, 3, 4, 5) and Cases (6, 7).*

**Response and revisions:**

We thank the reviewer's reminder. As suggested by the reviewer, we added the introduction of different cases **in Table 3 in the revised manuscript**.

**References**

Fu, T. M., Cao, J. J., Zhang, X. Y., Lee, S. C., Zhang, Q., Han, Y. M., Qu, W. J., Han, Z., Zhang, R., Wang, Y. X., Chen, D., and Henze, D. K.: Carbonaceous aerosols in China: top-down constraints on primary sources and estimation of secondary contribution, Atmospheric Chemistry and Physics, 12, 2725-2746, 10.5194/acp-12-2725-2012, 2012.

Chang, X., Wang, S., Zhao, B., Cai, S., and Hao, J.: Assessment of inter-city transport of particulate matter in the Beijing-Tianjin-Hebei region, Atmospheric Chemistry and Physics, 18, 4843-4858, 10.5194/acp-18-4843-2018, 2018.

Trang, T., Huy, T., Mansfield, M., Lyman, S., and Crosman, E.: Four dimensional data assimilation (FDDA) impacts on WRF performance in simulating inversion layer structure and distributions of CMAQ-simulated winter ozone concentrations in Uintah Basin, Atmospheric Environment, 177, 75-92, 10.1016/j.atmosenv.2018.01.012, 2018.

Ran, L., Pleim, J., Gilliam, R., Binkowski, F. S., Hogrefe, C., and Band, L.: Improved meteorology from an updated WRF/CMAQ modeling system with MODIS vegetation and albedo, Journal of Geophysical Research-Atmospheres, 121, 2393-2415, 10.1002/2015jd024406, 2016.

**Reviewer #2**

*0. In this study, Zhao et al. uses ground-based elemental carbon (EC) measurements from two sites in eastern China to evaluate and constrain black carbon (BC) emissions from two bottom-up inventories: a national/regional inventory for China (MEIC) and a high-resolution inventory for city clusters in southern Jiangsu Province. Both inventories include emissions from transportation, industry, power generation, and the residential sector. The authors show that the posterior emission estimates, constrained by ground measurements, are much smaller than the prior emission estimates, suggesting that pollution control measures by the Jiangsu government have effectively reduced emissions of BC. They also show results from various sensitivity tests, including those on the number of observation sites, spatial representativeness of observation sites, a priori emission inventories, and wet deposition. Overall, this is an interesting study that can be potentially useful for air quality modeling and management, emission inventory development and evaluation, and also studies on regional aerosol effects. Through several fairly detailed sensitivity tests, the authors also demonstrate that the differences between a priori and posteriori emission estimates are robust. However, the paper is overly long (and needs some improvement in presentation quality) and some reorganization may help. And there are also some concerns about the methodology that need to be addressed before this paper can be published in ACP.*

**Response and revisions:**

We appreciate the reviewer's remarks on the importance of the work. We reorganized Figures and Tables following the reviewer's suggestions (please see our response to Q3 and Q7) and specified the methodology of top-down estimate (please see our response to Q1-2). Please see the details in the following response and revision list to the reviewer's comment.

***1.*** *Major comments: It is not quite clear whether emissions outside of Jiangsu Province (but within the model domains) are scaled or not. Given the location of the sites, they could be strongly influenced by emissions from nearby provinces. If different local governments implemented different pollution control measures but the same domain wide scaling factors are used for emissions, that may lead to biases in the final estimated emissions for southern Jiangsu.*

**Response and revisions:**

We appreciate the reviewer's important comment. For MEIC-prior and JS-prior, emissions from different provinces and cities within the modeling domain were scaled based mainly on changes in their respective activity levels from 2012 to 2015, including those outside of Jiangsu Province. However, we did not constrain the emissions outside of Jiangsu Province in the top-down method, and we agree the limitation here. The main reason is that there were very few BC observation data available in the cities outside southern Jiangsu. Using observations at NJU or PAES to constrain emissions from those cities would bring more uncertainty for the cases in which local emissions dominated the air quality. Given this limitation, therefore, more measurements with better spatial coverage were recommended to be conducted and published for constraining BC emissions effectively in the future. We discussed this **in lines 545-547 in the revised manuscript**.

The uncertainty of using observations at two sites to constrain emissions from southern Jiangsu was expected to be insignificant in this work. Located in the downwind of the Yangtze River Delta region (YRD), NJU is more representative for the emissions from western YRD through regional transport. PAES is in urban Nanjing and its air quality is commonly influenced by surrounding transportation and residential sources, thus PAES is representative for the local emissions of Nanjing. We quantified the contribution of Nanjing and Suzhou-Wuxi-Changzhou-Zhenjiang city cluster through the brute-force method **in Sector 4.1 in the revised manuscript**. As can be seen **in Figure S10 in the revised supplement**, the monthly mean contributions of the emissions from the two regions in April were aggregated at 54%

and 59% at NJU and PAES respectively. We thus believe it is reasonable to use observations at two sites to constrain emissions from southern Jiangsu.

Regarding the influence of emissions outside southern Jiangsu, the contribution of each sector ($c_{power}$, $c_{industry}$, $c_{residential}$, and $c_{transportation}$) **in Eq1 in the revised manuscript** was simulated when the emissions from that sector were zeroed out for the whole third domain. It means that the emissions outside southern Jiangsu were also considered in the multiple regression model to obtain scaling factors. We applied the scaling factors to constrain emissions from southern Jiangsu only while remaining emissions outside southern Jiangsu unchanged so that it could better quantify the improvement of modeling performance at two sites due to the top-down estimate in southern Jiangsu. We acknowledge the uncertainty of including emissions of the whole third domain in the multiple regression model, due to different implementation of pollution control measures by city. As shown in Table R2, we compared the reduction rates of monthly BC emissions in the national inventory MEIC from 2012 to 2015 inside and outside southern Jiangsu in the domain. The difference between the two regions was less than 6%, implying the similar progress of pollution control measurements in two regions. Due to limited BC observations, moreover, we also checked the annual reduction rates in $PM_{2.5}$ concentrations from 2013 to 2015 for cities in the third domain based on the observation data from China National Environmental Monitoring Center (http://www.cnemc.cn/). As shown in Table R3, the annual reduction rates were ranged from 10% to 17% by city, reflecting again the similar implementation of air pollution control policies around the regions. Relative statement was added **in lines 222-235 in the revised manuscript**, and Tables R2 and R3 were included as **Tables S2 and S3 in the revised supplement**.

Table R2. Reduction rates in monthly emissions from 2012 to 2015 in MEIC for southern Jiangsu and other regions within the third modeling domain.

| Region | Jan. | Apr. | Jul. | Oct. |
|---|---|---|---|---|
| Southern Jiangsu (%) | 18 | 18 | 26 | 21 |
| Outside southern Jiangsu (%) | 12 | 16 | 21 | 15 |

Table R3. Reduction rates in annual PM$_{2.5}$ concentration for cities within the third modeling domain from 2013 to 2015.

| Province | City | Reduction rate (%) |
|---|---|---|
| Anhui | Hefei | 15.26 |
| | Nantong | 15.90 |
| | Taizhou | 11.76 |
| | Yangzhou | 16.84 |
| Jiangsu | Nanjing | 15.58 |
| | Suzhou | 12.76 |
| | Wuxi | 10.45 |
| | Changzhou | 12.31 |
| | Zhenjiang | 12.80 |
| Shanghai | Shanghai | 10.88 |

*2. The lack of biomass burning emissions can be concerning. Could the model underestimates of BC in July and particularly October be caused by the biomass burning (particularly agricultural fires)? How does the lack of biomass burning emissions affect the estimated emissions for other sectors?*

**Response and revisions:**

We thank the reviewer's comment. In both inventories (MEIC and JS), the emissions came from four sectors, including power generation, industry, residential sources and transportation, and the residential sources included fossil fuel and biofuel combustion. However, we did not include emissions from biomass open burning. In another paper of our group (Yang and Zhao, 2019), the emissions from biomass open burning in YRD were thoroughly evaluated with various methods, and the emissions were estimated to decrease by 60% from 2012 to 2015 in southern Jiangsu attributed mainly to the enhanced control of crop burning activities by the local government. With the optimized constrained method, the BC emissions from crop open burning were calculated at 0.83 Gg in southern Jiangsu 2015, contributing small in the JS-prior and JS-posterior at 3% and 6%, respectively. As shown in Table R4, in addition, the most intensive crop burning was found in May and August, indicated by the monthly fire points from satellite dectection. Limited effect of biomass burning was thus expected for the modeling periods in this study.

Table R4. Monthly fire points in southern Jiangsu for 2015, taken from Moderate Resolution Imaging Spectroradiometer (MODIS) Global Monthly Fire Location Product (MCD14ML).

| 2015 | Jan. | Feb. | Mar. | Apr. | May | Jun. | Jul. | Aug. | Sep. | Oct. | Nov. | Dec. |
|------|------|------|------|------|-----|------|------|------|------|------|------|------|
| Fire point | 9 | 11 | 12 | 58 | 249 | 30 | 96 | 127 | 16 | 9 | 1 | 10 |

In this work, the scaling factor of residential sources in October was estimated at 1.52 in JS-posterior, implying the enhancement in BC emissions in autumn to JS-prior. The result thus implied that there were missing sources likely associated with crop waste burning in autumn, and it was discussed **in lines 420-424 in the revised manuscript**. We also evaluated the sensitivity of the constraining method to the initial emission input **in Section 4.2 in the revised manuscript**, and found the uncertainty from the a priori inventory had limited effects on the top-down estimate. To summarize, therefore, we believe that lack of biomass burning emissions in the initial inventories would not significantly bias the top-down estimation.

*3.* *The paper is overly long and can be better organized. In particular, if spatial representativeness and wet deposition are important, can the authors focus on the top-down estimates that consider both of these factors? Description of the other sensitivity tests can be brief. Also writing needs to be improved.*

**Response and revisions:**

We thank the reviewer's comment. To make the manuscript concise, we moved Figures 9 and 10 in the original manuscript to the revised supplement (Figures S11 and S12) given that the near-linearity was also indicated in previous studies (Wang et al., 2013). We integrated the original Table 8 into Table 3 in the revised manuscript to summarize the modeling performances of different cases. The scaling factors and statistical indicators in Case 7 in the original Table 9 were integrated into Table 5, while emissions by sector in Case 7 and the relative deviations compared to JS-posterior in Table 9 were integrated into Table 6. We moved the original Figure 3 that presents the seasonal variations in emissions of JS-prior, JS-posterior and MEIC-prior to the revised supplement (the new Figure S8) given the less statistical significant in seasonal patterns of several sectors in JS-posterior. We also moved the original Table 5 that summarizes the emissions from Nanjing and other cities in southern Jiangsu in different cases to Table S9 in the revised supplement. Sections 4.1 and 4.2 in the original manuscript were merged into one section (Section 4.1 in the revised manuscript) to evaluate the effects of number and spatial representativeness of observation sites on the top-down estimate. We believe the analysis on the uncertainty of the a priori inventory was important, as it could help judge the robustness of the constraining method. We found the influence of the a priori emissions was limited, and implied that the method could be potentially applied even if uncertainty existed in the bottom-up inventory. Therefore, we kept this part in the revised manuscript.

*4. Specific comments: Figure 3 and the paragraph starting from line 389: given that the scaling factor for April and Oct. are more uncertain (in terms of their statistical significance), are the seasonal patterns in the posterior emission estimates significant?*

**Response and revisions:**

We thank and agree with the reviewer's comment. Though the multiple regression model was statistically significant as a whole indicated by 0.00 of the overall significance in four months, the estimates for certain sources including industry in April and October and residential in April and July were more uncertain to some extent, as illustrated **in Table 1 in the revised manuscript**. It implied that the constrained emissions for those months/sources need to be cautiously applied in CTM and the seasonal patterns in those sectors could be less significant. Relevant discussion was **in lines 383-386 in the revised manuscript** and we moved original Figure 3 that presents the seasonal variations in emissions of JS-prior, JS-posterior and MEIC-prior to **Figure S8 in the revised supplement**.

**5.** *Figure 5a – what may have caused the model overestimates in mid-January at PAES? How does this period affect emission estimates? Can the authors exclude this period and compare the top-down estimates?*

**Response and revisions:**

We thank the reviewer's comment. The overestimation in January at PAES (especially in middle and late January, 16$^{th}$–26$^{th}$) may result from the emission control policy implemented for the National Memorial Day of Nanjing Massacre Victims in December 13$^{th}$ in 2014. During the period, Nanjing was undertaking series of stringent restrictions on air pollutant emissions. For example, key petrochemical and steel industries were shut down, and all the high-pollution vehicles were forbidden to drive in Nanjing. Those restrictions had large impacts on emissions and thereby air quality in the following month at PAES, but have not been fully considered in current emission inventories. Beside the emission control measures implemented in Nanjing, we evaluated the effect of planetary boundary layer (PBL) height on the modeling performance at PAES, as illustrated in Figure R1. Higher daily average PBL height was found for periods when the simulated concentrations were relatively lower (e.g., 6$^{th}$ -7$^{th}$, 12$^{th}$-15$^{th}$ and 28$^{th}$-31$^{st}$), resulting in smaller bias between simulations and observations. In contrast, the lower PBL height found in other periods would exaggerate the overestimation in simulated concentrations, given the elevated emissions in JS-prior. We added the analysis **in lines 454-468 in the revised manuscript** and included Figure R1 as **Figure S9 in the revised supplement**. Attributed to the instrument maintenance, moreover, the observation data in January at PAES were relatively insufficient, and the data points were 70% less than those at NJU. Therefore, the contribution of observation at PAES was limited in the multiple regression model.

[Figure]

Figure R1. The simulated daily average PBL heights in January 2015 at PAES.

Following the reviewer's suggestion, we excluded the data points in middle and late January ($16^{th}$ -$26^{th}$) at PAES and re-compared the observed and simulated BC concentrations. As shown in Table R5, the overestimation in CTM was largely reduced when the data were excluded, and the top-down estimate corrected the bias moderately at PAES. We added the discussions **in lines 499-503 in the revised manuscript** and added Table R5 as **Table S6 in the revised supplement**.

Table R5. Statistical indicators for observed and simulated BC concentrations using JS-prior and JS-posterior in January excluding data from $16^{th}$ to $26^{th}$ at PAES.

| Site | Parameter | JS-prior | JS-posterior |
|------|-----------|----------|--------------|
| | Average SIM ($\mu g/m^3$) | 2.86 | 2.68 |
| | Average OBS ($\mu g/m^3$) | 2.15 | 2.15 |
| PAES | NMB (%) | 32.95 | 24.65 |
| | NME (%) | 52.61 | 49.63 |
| | R | 0.72 | 0.74 |

*6.* *Lines 456-461: again, could the model bias be due to the lack of biomass burning emissions?*

**Response and revisions:**

We thank the reviewer's comment. The bigger bias found in July and October at NJU when applying JS-posterior resulted mainly from the limitation of the constraining method. We used observations at two sites to constrain emissions from southern Jiangsu as a whole. Therefore, overestimation and underestimation in concentrations at different sites could not be corrected simultaneously without considering the spatial representation of observation sites, as discussed **in lines 511-516 in the revised manuscript**.

The underestimation in BC concentrations for July and October with JS-prior could be partly due to the lack of biomass open burning emissions. However, such influence was expected to be insignificant (please see our response to Q2), and the impact of the a priori emission input was found limited on the top-down estimation, as discussed **in Section 4.2 in the revised manuscript**.

**7.** *Tables: There are already many tables in the paper (and maybe not everyone is absolutely necessary). But a table that summarizes the different cases may be helpful for readers to keep track.*

**Response and revisions:**

We thank and follow the reviewer's comment to make the tables concise. We integrated the original Table 8 to a new Table 3 in the revised manuscript to summarize the modeling performance for different cases. For the original Table 9, moreover, the scaling factors and statistical indicators from the multiple regression model in Case 7 were integrated to Table 5, and the emissions by sector and the relative deviations to JS-posterior in Case 7 were integrated to Table 6. We also moved the original Table 5 that summarizes the emissions from Nanjing and other cities in southern Jiangsu in different cases to Table S9 in the revised supplement.

**8.** *Table 4 and related discussion on case 3: would the authors expect somewhat different driving conditions and emission factors for automobiles in urban and*

*suburban settings? If so, is it still a valid assumption to assume the same scaling factor between NJU and PAES for transportation?*

**Response and revisions:**

We thank the reviewer's comment. In Case 3, we assumed a same scaling factor for transportation for different cities in southern Jiangsu to avoid the collinearity in the multiple regression model. As the observation data at NJU and PAES were applied to constrain emissions from Suzhou-Wuxi-Changzhou-Zhenjiang city cluster and Nanjing, respectively, the assumption of a same scaling factor at NJU and PAES did not mainly indicate the similar driving conditions or emission factors for automobiles in suburban and urban. Instead, it mainly indicated that the relative changes in emissions from transportation were similar across the cities in southern Jiangsu from 2012 to 2015. As we stated **in lines 591-593 in the revised manuscript**, such assumption is expected to be reasonable, because of the same progress of emission standard implementation (National Standard Stage IV) in southern Jiangsu and the frequent circulation of vehicles among the cities.

**References**

Wang, X., Wang, Y., Hao, J., Kondo, Y., Irwin, M., Munger, J. W., and Zhao, Y.: Top-down estimate of China's black carbon emissions using surface observations: Sensitivity to observation representativeness and transport model error, Journal of Geophysical Research: Atmospheres, 118, 5781-5795, 10.1002/jgrd.50397, 2013.

Yang, Y., and Zhao, Y.: Quantification and evaluation of atmospheric pollutant emissions from open biomass burning with multiple methods: a case study for the Yangtze River Delta region, China, Atmospheric Chemistry and Physics, 19, 327-348, 10.5194/acp-19-327-2019, 2019.

**Reviewer #3**

*0. This is a very nice and detailed work to constrain BC emissions in southern Jiangsu. The approach and uncertainty analysis may be applied to other regions. The paper is well written in general, suitable for ACP. Below are a few suggestions to further improve the paper.*

**Response and revisions:**

We appreciate the reviewer's positive remarks on the importance of the work. Please see the details in the following response and revision list to reviewer's comment.

*1. It would be nice to discuss in the conclusion section the potential of applying the method to other regions.*

**Response and revisions:**

We thank the reviewer's comment. The method could be applied to constrain the BC emissions for other regions effectively if there are sufficient observation data with satisfying spatiotemporal coverage. We added the statement **in lines 796-799 in the revised manuscript**.

*2. The regression model needs to be further clarified. Are the scaling factors (betta) for each month, day, or hour? Why is there not a term in Eq. 1 for the background (e.g., lateral boundary condition) reflecting the effect of horizontal transport from regions other than southern Jiangsu? Table S3 and Fig. S7 show that the sum of southern Jiangsu contributions is much smaller than 100%, implying a large contribution from regions other than southern Jiangsu.*

**Response and revisions:**

We thank the reviewer's comment. The scaling factors were obtained for each month and used to constrain the monthly emissions in southern Jiangsu. We clarified it **in lines 235-237 in the revised manuscript**.

Regarding the background reflecting the regional transport, $c_{power}$, $c_{industry}$, $c_{residential}$ and $c_{transportation}$ in the multiple regression model were simulated by brute-force method in CTM in which emissions from corresponding sector in the third domain were zeroed out. Therefore the contributions of emissions outside southern Jiangsu in the third domain were considered in the model. Moreover, $\varepsilon$ reflected the effect of background conditions (e.g., emissions in the first and second domain in CTM and emissions not included in the a priori inventory like those from natural sources). We clarified it **in lines 222-227 and 237-239 in the revised manuscript**. For example, the $\varepsilon$ was estimated at 0.96 μg/m$^3$ in the multiple regression model for April in JS-posterior. By zeroing out the emissions from the third domain in CTM, the monthly contribution from boundary conditions were calculated at 0.76 and 0.77 μg/m$^3$ at NJU and PAES, respectively. In spite of the modest bias between $\varepsilon$ and the estimated contribution of boundary conditions, including $\varepsilon$ would reduce the uncertainty of the multiple regression model.

We added the contributions from four sectors in the third domain at the two sites **in Table S5 in the revised supplement**. The total contributions were larger than 50% for all the months and sites except for January. We assumed that the smaller contributions in January resulted partly from the longer lifetime of BC in winter due to less wet deposition. We also identified the transport pathways of air masses sampled at NJU for the four months through cluster analysis of back trajectories with Hybrid Single Particle Lagrangian Integrated Trajectory (HYSPLIT, version 4) model as illustrated in Figures R2. Compared to other months, fewer air masses passed through the third modeling domain in January due to the prevailing northerly wind, implying more contribution from regional transport to the air quality at the site in January. Similar results were found for other region. Jia et al. (2008) estimated that regional transport on average contributed nearly 50% of PM (up to 70% in southerly regions) in winter in three sites in Beijing. Sun et al. (2014) considered the accumulation of local BC emissions and estimated a contribution of 53% from regional transport to BC in Beijing. Given the smaller contribution of emissions within the third domain in January, we acknowledged that the multiple regression model was less effective on identifying the sources of BC in winter by constraining the emissions in southern Jiangsu city cluster alone. We added the discussion **in lines 360-370 in the revised manuscript** and included Figure R2 as **Figure S6 in the revised supplement**.

[Figure]

(a)

(b)

(c)

(d)

Figure R2. The transport pathways of air masses sampled at NJU based on cluster analysis of back trajectories in HYSPLIT model in January (a), April (b), July (c) and October (d).

**3.** *The idea of testing the spatial representativeness of measurements is very nice. Given the spatial representativeness difference between the two sites, is it possible to use Case 3 as your best case? Alternatively, it would be nice to improve the regression model by taking into account the transport path, e.g., by basing on WRF modeled winds to design a model that considers the trajectory of air movement. The much higher bias in JS-posterior than JS-prior in Case 1, which is a concern, is related to this spatial representativeness issue.*

**Response and revisions:**

We thank the reviewer's comment. Among all the cases discussed in the paper, the best CTM performance was obtained in Case 3 in which observations at both sites were used with their difference in spatial representativeness considered in the constraining method. We also appreciate the reviewer's suggestion, which could potentially improve the analysis of spatial representativeness and could be applied with more observation data available in the future. The larger NMEs in July and October at NJU in JS–posterior than JS-prior were related to the spatial representativeness issue, which was discussed **in lines 511-516 in the revised manuscript**.

**4.** *A clearer discussion of temporal resolution in bottom up inventories and how this resolution affects the top-down constraint will be very helpful.*

**Response and revisions:**

We thank the reviewer's comment. We derived the hourly bottom-up emission inventory for CTM. The monthly distributions of emissions from power plants and industry plants in JS-prior were dependent on those of electricity generation and typical industrial production, respectively. Such information was investigated by Zhou et al. (2017) according to the official statistics of the country (http://data.stats.gov.cn/). Meanwhile, the real-time monitoring on urban traffic in Nanjing was applied to allocate the temporal distribution of emissions from on-road vehicles in the whole regions in JS-prior. The weekly and hourly distributions of different sources in YRD (Li et al., 2011) were adopted to further allocate emissions in JS-prior. For MEIC-prior, we obtained the monthly emissions directly and applied the same weekly and hourly distributions as JS-prior. We described this **in lines 207-215 in the revised manuscript**. The temporal distributions based on local statistical data were expected to be more reliable in CTM than other information. Regarding the effect of the monthly variation on the constraint method, we compared top-down estimate derived from JS-prior and MEIC-prior in April, respectively, **in Section 4.2 in the revised manuscript**. Similar emission estimation, spatial distribution and modeling performance were found for the two a posteriori emissions, even clear difference existed in the two a priori inventories. The result thus implied the insignificant effect of monthly variation of emissions on the top-down constraint. We discussed this **in lines 667-671 in the revised manuscript**. We did not constrain the hourly emissions in this study and the hourly distribution was thus unchanged in the top-down estimate.

*5. Comparison with near-surface measurements is sensitive to WRF/CMAQ modeled vertical processes, including the number of vertical layers within the PBL, the thickness of the first layer, and the model error in vertical mixing representation. WRF/CMAQ may have some issues with PBL mixing (Liu et al., 2018). Please specify these model setups. Please discuss the potential effect of model vertical resolution/mixing/transport errors on the BC constraint.*

**Response and revisions:**

We thank the reviewer's comment. The PBL module adopted in WRF 3.4 was ACM2, and the information was added **in line 285 in the revised manuscript**. There were 27 vertical layers in the model, with the heights of 54, 132, 234, 362, 523, 729, 974, 1417, 1887, 2385, 2914, 3900, 4890, 5886, 6885, 7885, 8891, 9907, 10946, 12000, 13070, 14158, 15278, 16441, 17662, 18966 and 20405 m, respectively. The simulated monthly average PBL heights along with the range of hourly simulations at

NJU and PAES in four months were shown in Table R6. Therefore, there were average 5 vertical layers within the PBL. We found the similar result of the low simulated PBL height in WRF/CMAQ model as Liu et al. (2018) and the overestimation of BC concentration at PAES even after top-down constraint may result from it. We added the analysis **in lines 503-507 in the revised manuscript** and included Table R6 as **Table S7 in the revised supplement**.

The effect of vertical distribution on BC emission constraining was evaluated for Asia by Zhang et al. (2015). They repeated the top-down inversions using the OMI retrieval absorption aerosol optical depth (AAOD) based on the CALIOP and GOCART aerosol layer height and found the difference in the optimized BC emissions were less than 30% in April and 10% in October compared to the optimized emissions using the initial GEOS-Chem model. The difference was within the acceptable range compared with up to 500% enhancements in April and 10-50% in October with the top-down constraining. When applying ground observations in this study rather than column concentration in AAOD, the effect of vertical distribution could be smaller.

Table R6. The simulated monthly average PBL heights and the range of hourly simulations at NJU and PAES in four months.

| Month | Site | Monthly average PBL (m) | Hourly average PBL (m) |
|---|---|---|---|
| January | NJU | 370.25 | 27.59-1443.64 |
| | PAES | 384.56 | 27.20-1460.07 |
| April | NJU | 432.73 | 28.61-2157.87 |
| | PAES | 441.72 | 28.61-2157.87 |
| July | NJU | 381.14 | 30.70-1617.69 |
| | PAES | 431.02 | 30.02-1975.01 |
| October | NJU | 462.57 | 29.70-2065.97 |
| | PAES | 488.30 | 29.78-2073.46 |

**6.** *Table S2 shows that the prevailing winds in all three meteorological sites are southerly or southeasterly. I thought there would be northerly in the cold months (January and October). Please double check.*

**Response and revisions:**

We thank the reviewer's comment. We checked the simulated and observed wind directions again and found the same result. The NMEs of wind directions were found below 40% at three meteorological stations in January and October, reflecting the robustness of the WRF modeling. In January, the average simulations and observations **in Table S4 in the revised supplement** did not mean that the prevailing winds were southerly. The values were the mean of the northerly wind directions ranging from 0-45° or 315-360°. Taking the wind directions at Hongqiao in January and October as examples, the prevailing winds were northerly and easterly in winter and autumn, respectively, as shown in Figures R3.

[Figure]

(a)                                                    (b)

Figure R3. Wind speeds and directions at Hongqiao in January (a) and October (b).

**7.** *Some paragraphs are too long and should be splitted, for example, L71-111, L352-388.*

**Response and revisions:**

We thank the reviewer's comment. As suggested by the reviewer, we split L71-111 in the initial manuscript into two parts, one was about the large uncertainties in bottom-up emission inventories, and the other was the challenge existing in updating BC inventories continuously, **in lines 74-114 in the revised manuscript**. We split L352-388 in the original manuscript and reorganized the paragraphs. One was about the relative change between JS-prior and JS-posterior, and the other was the detailed description about scaling factors for different sectors, **in lines 387-424 in the revised manuscript**.

**8.** *Abstract – please specify that monthly, sector-level and city-level emissions are optimized.*

**Response and revisions:**

We thank the reviewer's comment. We followed the suggestion and specified the optimized monthly, sector-level and city-level emissions in **in line 24 in the revised manuscript**.

**9.** *L22 – "observations," should be "observations" (no comma).*

**Response and revisions:**

We thank the reviewer's reminder and deleted the comma **in line 23 in the revised manuscript**.

**10.** *Abstract – please specify that WRF/CMAQ is used.*

**Response and revisions:**

We thank the reviewer's reminder and specified the WRF/CMAQ model **in lines 21-22 in the revised manuscript**.

**11.** *L214 – is there is term for background (due to horizontal transport)?*

**Response and revisions:**

We thank the reviewer's comment. $\varepsilon$ reflected the effect of emissions from background conditions, which was added **in lines 237-239 in the revised manuscript**. (please also see our response to Q2).

**12.** *L218 – "domain-wide" – here you optimize the southern Jiangsu emissions, not the domain-wide emissions. Also, as suggested above, an improved regression model may be used to better account for spatial representativeness of measurements.*

**Response and revisions:**

We thank the reviewer's comment and revised the words for $\beta_1$-$\beta_4$ **in lines 235-237 in the revised manuscript**. We appreciate the reviewer's suggestion to improve the multiple regression model and it could be applied with more observation data available in the future to better consider spatial representativeness.

**13.** *L256 – "coordinated" should be "coordinate"*

**Response and revisions:**

We thank the reviewer's reminder and corrected the word **in line 281 in the revised manuscript**.

**14.** *L274-288 – please specify the temporal resolution of bottom up emissions.*

**Response and revisions:**

We thank the reviewer's comment. We specified the temporal distributions of two bottom-up emission inventories used in CTM **in lines 207-215 in the revised manuscript**. The monthly distributions of emissions from power plants and industry plants in JS-prior were dependent on those of electricity generation and typical industrial production, respectively. Such information was investigated by Zhou et al. (2017) according to the official statistics of the country (http://data.stats.gov.cn/). The real-time monitoring on urban traffic in Nanjing was applied to allocate the temporal distribution of emissions from on-road vehicles in the whole regions in JS-prior. The weekly and hourly distributions of different sources in the Yangtze River Delta (Li et al., 2011) were directly adopted to further allocate the emissions in JS-prior. For MEIC-prior, we obtained the monthly emissions and applied the same weekly and hourly distributions as JS-prior. The temporal allocations based on local statistical data were expected to be more reliable in CTM.

**15.** *L283-285 – do you remove emissions in the whole domain, or just southern Jiangsu cities?*

**Response and revisions:**

 We thank the reviewer's comment. We removed emissions in the whole third domain, and it was specified **in lines 308-310 in the revised manuscript**.

**16.** *L288 – "Scenarios B and S" should be "Scenarios B and S1-S4"*

**Response and revisions:**

 We thank the reviewer's reminder and revised it **in line 313 in the revised manuscript**.

**17.** *L324 – "double" should be "twice"*

**Response and revisions:**

 We thank the reviewer's comment and corrected the word **in line 349 in the revised manuscript**.

**18.** *L340 – "VIF smaller than 10" – the VIF values in the table are much smaller than 10.*

**Response and revisions:**

 We thank the reviewer's reminder and revised it **in lines 374-376 in the revised manuscript**.

**19.** *L386-388 – this sentence is not clear*

**Response and revisions:**

We thank the reviewer's comment. Based on the bottom-up approach, Huang et al. (in preparation) incorporated detailed information and changes of individual sources, and estimated BC emissions for Nanjing from 2012 to 2015. The emissions in 2015 were estimated to decrease by 60% compared to those in 2012, and this relative change was close to that for the southern Jiangsu (a 50% reduction from JS-prior to JS-posterior) found in this study. The top-down method could thus capture the changes in emissions due to improved control measures. We revised the sentence **in lines 395-398 in the revised manuscript**.

**20.** *L418-442 – A figure would be much better than a table for this type of analysis.*

**Response and revisions:**

We thank the reviewer's comment. **Figures 3 and 4 in the revised manuscript** illustrated the simulated BC concentrations based on JS-prior and observations in four months at NJU and PAES, respectively. The analysis mentioned by the reviewer was reflected in those figures.

**21.** *L426 – what do you mean by "commonly"? The wording may be improved.*

**Response and revisions:**

We thank the reviewer's reminder and replaced the word commonly with generally **in line 472 in the revised manuscript**.

**22.** *L443-446 – The increased bias from JS-prior to JS-posterior at NJU should be discussed in more detail.*

**Response and revisions:**

We thank the reviewer's comment. The increased bias from JS-prior to JS-posterior in July and October at NJU and the detailed analysis was mentioned **in lines 508-516 in the revised manuscript**. It resulted mainly from the limitation of current multiple regression model that overestimation and underestimation in concentrations at different sites could hardly be corrected simultaneously without further improvement in spatial distribution of emissions.

**23.** *L464 – some cases are for other months.*

**Response and revisions:**

We thank the reviewer's comment. The sensitivities to observation and bottom-up emission input were evaluated in April (Cases 2-5). We evaluated the near linearity between emissions and concentrations in July and October as the two months were identified as the months with the most and least impact from precipitation suggested by simulated wet deposition to emission ratio. The impacts of simulated wet deposition and satellite-derived accumulated precipitation on top-down estimate were evaluated in July (Case 6-7). We had specified it **in lines 518-525 in the revised manuscript**.

**24.** *L551 – "initial" should be "a priori". Please revise throughout the text.*

**Response and revisions:**

We thank the reviewer's reminder and revised it throughout the text.

**25.** *L573-604 – the paragraph contains multiple messages, and is better to be splitted.*

**Response and revisions:**

We thank the reviewer's comment. As suggested, the smaller difference in BC emissions and simulated concentrations between JS-posterior and MEIC-posterior were split **in lines 639-666 in the revised manuscript**. The effect of the a priori bottom-up emission inventories on top-down estimate was summarized in another paragraph **in lines 667-671 in the revised manuscript**.

**26.** *Figs. S8-11 – the dates of precipitation are also not very well simulated.*

**Response and revisions:**

We thank the reviewer's comment and delete the evaluation of simulated precipitation dates **in lines 703-704 in the revised manuscript**. Considering the large discrepancy between simulated and observed precipitation, we conducted Case 7 to screen satellite-derived precipitation and compared the top-down estimates in two cases.

**27.** *L701 – "insignificant" should be "modest"*

**Response and revisions:**

We thank the reviewer's reminder and revised it **in line 763 in the revised manuscript**.

**28.** *L715-717 – the increased bias at NJU should be mentioned.*

**Response and revisions:**

We thank the reviewer's comment and mentioned the increased bias **in lines 779-780 in the revised manuscript**.

**29.** *L735-737 – it would be extremely difficult to use satellite AOD to constrain BC emissions.*

**Response and revisions:**

We thank the reviewer's comment and deleted the texts in the revised manuscript.

**Abstract**

We combined a chemistry transport model (the Weather Research and Forecasting and the Models-3 Community Multi-scale Air Quality Model, WRF/CMAQ), a multiple regression model and available ground observations to optimize  black carbon (BC) emissions at monthly, emission sector and city cluster level. We derived top-down emissions and  
[revised manuscript text omitted]

input. The near-linearity assumption in uncertainties of the multiple regression model
and the effect of wet deposition on the top-down estimate were finally evaluated
including the influence of precipitation and the near linear assumption between BC

emissions and concentrations.

**2 Data and method**

**2.1 Bottom-up inventories of BC emissions**

Two bottom-up emission inventories at different spatial scales were used in this work. At the national scale, the Multi-resolution Emission Inventory for China (MEIC, http://www.meicmodel.org/) was developed by Tsinghua University, with an original horizontal resolution at $0.25° \times 0.25°$. At the provincial scale, Zhou et al. (2017)

collected the best available information of industrial sources in Jiangsu and developed an inventory with higher resolution at $3 \times 3$ km. The latter was proved to be more supportive in air quality simulation at city cluster scale (Zhou et al., 2017; Zhao et al.,

2017). In both inventories, anthropogenic BC emissions for 2012 came from four major sectors: power generation, industry, residential sources and transportation. The national and provincial inventories for 2015 (mentioned respectively as MEIC-prior and JS-prior hereinafter) were obtained using a simple scaling method based mainly on changes in activity levels (energy consumption and industrial production, etc)

between the four years. Table S1 in the supplement summarizes the data sources of activity levels and the scaling factors by sector in JS-prior. As MEIC-prior includes only four major sectors, the scaling factor for each sector was calculated as the average of those for subcategories within the sector. Potential changes in BC emission factors from 2012 to 2015, e.g., those attributed to varied manufacturing technologies and/or penetrations of emission control devices, were not considered in the calculation.

The implication and uncertainty from that simplified emission scaling method will be further discussed in Section 4.32. The temporal distribution of the emissions was dependent on that of activity levels by source category.The monthly distributions of emissions from power plants and industry plants in JS-prior were dependent on those of electricity generation and typical industrial production, respectively. Such information was investigated by Zhou et al. (2017) according to the official statistics of the country (http://data.stats.gov.cn/).

The real-time monitoring on urban traffic in Nanjing was applied to allocate the temporal distribution of emissions from on-road vehicles in the whole regions in

JS-prior. The weekly and hourly distributions of other sources were taken from  Yangtze River Delta (Li et al., 2011).

For MEIC-prior, we obtained the monthly emissions directly and  applied the same weekly and hourly distributions as

JS-prior.

**2.2 Top-down emission estimation with multiple regression model**

The top-down emissions of BC in southern Jiangsu (mentioned as JS-posterior hereinafter) were estimated with a multiple regression model using ground observations as constraint. The regression model matched BC contributions by sector (calculated through CTM) against measured ambient hourly BC concentrations:

$$c_{obs} = \beta_1 c_{power} + \beta_2 c_{industry} + \beta_3 c_{residential} + \beta_4 c_{transportation} + \varepsilon \tag{1}$$

where $c_{obs}$ is the vector of observed hourly BC concentrations. $c_{power}$, $c_{industry}$, $c_{residential}$, and $c_{transportation}$ are the vectors of BC concentrations contributed by power generation, industry, residential sources and transportation in southern Jiangsu and nearby  regions (the third domain of air quality modeling, as described later in

Section 2.3), respectively, and they were simulated using the brute-force method as described in Section 2.3. Southern Jiangsu and nearby cities were considered as a whole in the multiple regression model  based on an assumption of similar  implement of air pollution control measures  for the two regions.

Tables S2

and S3 in the supplement summarize respectively  the reduction rates in BC emissions estimated by  MEIC and those in observed $_{2.5}$ concentrations for  recent years for southern Jiangsu and nearby cities   The discrepancies in reduction rates between the two regions were found less than 6% and 7% for monthly BC emissions and annual PM$_{2.5}$ concentrations, respectively, implying the similar progress of emission control and air quality improvement.  $\beta_1$-$\beta_4$ are the  scaling factors obtained by sector in the multiple regression model and were applied to optimize southern Jiangsu emissions to best match observations for each month.  $\varepsilon$ is the error vector of the model, reflecting the effect of background conditions (e.g., emissions  outside the third domain in CTM and emissions not included in the a priori inventory like those from natural sources). Before applying observations and simulated contributions by sector in the multiple regression model,  data screening was conducted following the criterions: the periods lack of observation data, those for which the contribution of each emission sector was simulated to be smaller than zero through the brute-force method , and those for which the sum of contributions of all the four sectors was larger than 100% . The data screening helped to reduce the uncertainty of CTM in the multiple regression model.

As BC is not one of the six regulated air pollutants in the NAAQS, it was a big challenge to obtain observation data with high temporal resolution in most cities of southern Jiangsu. For the whole year 2015, hourly ambient BC concentrations were available at two sites in Nanjing, the capital of Jiangsu. As illustrated in Figure 1, one is a suburban site located in the Xianlin Campus of Nanjing University in northeast Nanjing (NJU), and the other is an urban site in Jiangsu Provincial Academy of Environmental Science (PAES). At both sites, BC was sampled and analyzed hourly with semi-continuous carbon analyzer (Model-4, Sunset Lab, USA). Details of the measurement approach were described in Chen et al. (2017). The statistics of observed ambient BC concentrations at the two sites are shown in Figure S2 in the supplement. The annual average BC concentrations (calculated as the mean of January,

April, July and October) were 3.83 and 2.47 μg/m$^3$ at NJU and PAES, respectively.

The hourly average BC observations ranged 0.06-17.65 μg/m$^3$ and 0.22-19.76 μg/m$^3$

at NJU and PAES, respectively. The values were similar to those observed in the

Guanzhong basin (0.4-23.1μg/m$^3$), the Pearl River Delta region (1-13 μg/m$^3$) and the

Beijing-Tianjin-Hebei region (2-32 μg/m$^3$) (Li et al., 2016). Much higher BC

concentrations were observed in autumn and winter at both sites, with the monthly means at 3.96 and 5.44 μg/m$^3$ at NJU and 3.62 and 2.80 μg/m$^3$ at PAES, respectively.

The scaling factors derived from Eq. (1) were used to constrain BC emissions in southern Jiangsu in  –JS-prior from a top-down perspective by assuming a near-linear relation between changes in BC concentrations and emissions:

$E_{JS\text{-}posterior} = \beta_1 E_{power} + \beta_2 E_{industry} + \beta_3 E_{residential} + \beta_4 E_{transportation}$ (2)

where $E_{JS\text{-}posterior}$ is the vector of the total BC emissions from the top-down approach;

$E_{power}$, $E_{industry}$, $E_{residential}$ and $E_{transportation}$ are the vectors of BC emissions from power generation, industry, residential sources and transportation, respectively, in JS-prior.

**2.3 Air quality simulation**

We used the Models-3 Community Multi-scale Air Quality (CMAQ) version

4.7.1 to simulate ambient BC concentrations. As shown in Figure 1, three nested domains were applied with horizontal resolutions of 27, 9, and 3 km, respectively, on a Lambert Conformal Conic projection centered at (110°E, 34°N). The mother domain (D1, 177×127 cells) covered most parts of China and other surrounding countries. The second domain (D2, 118×121 cells) covered Jiangsu, Anhui, Zhejiang, Shanghai, and parts of other provinces in China–. The third domain (D3, 133×73 cells) covered

Shanghai, part of Anhui province and the city cluster in southern Jiangsu. There were

27 vertical levels from the ground surface up to 50 hPa on terrain-following coordinated. The simulations were conducted for January, April, July and October to represent four typical seasons in 2015. A 5-day spin-up period of each month was applied to minimize the influence of initial conditions in the simulations.

Meteorological fields were simulated by the Weather Research and Forecasting

Model (WRF) version 3.4. and ACM2 planetary boundary layer (PBL) mixing scheme, the carbon bond gas-phase mechanism (CB05) and AERO5 aerosol module were adopted in WRF/CMAQ model. Relevant details of model configuration can be found in Zhou et al. (2017). Statistical indicators including averages of simulations and observations, bias, normalized mean bias (NMB), normalized mean error (NME), root mean squared error (RMSE) and index of agreement (IOA) were applied to evaluate the modeling performance of WRF (Baker et al, 2004; Zhang et al., 2006).

Ground observation data at 1 or 3 h interval at meteorological stations including

Lukou, Hongqiao and Liyang stations in the third domain (labeled in Figure 1) were taken from National Climatic Data Center (NCDC). The statistical indicators for temperature at 2 m (T2) and relative humidity at 2 m (RH2), wind speed and direction at 10 m (WS10 and WD10) for the four typical months in 2015 are summarized in

Table S2 S4 in the supplement. Discrepancies between ground observations and WRF

modeling were within acceptable range (Emery et al., 2001).

To make it applicable in our CTM, MEIC-prior was downscaled into grid systems of each modeling domain, based on the spatial distributions of gross domestic product (GDP, for power generation and industrial emissions) and population (for residential and transportation emissions) at a horizontal resolution of 1×1 km. The downscaled MEIC-prior was used for the first, the second domains and the regions outside Jiangsu of the third domains, while JS-prior was applied for the Jiangsu region of the third domain. After applying the temporal distribution of the emissions discussed in Section 2.1, the hourly bottom-up emission inventories were used in

CTM. Compared with larger temporal resolution like monthly or annual, the hourly simulated concentrations generated from hourly emissions in CTM could evaluate the

批注 [zy2]: 但这些研究中的 hourly resolution 指的是观测浓度还是基于时变化排放模拟的浓度？

[revised manuscript text omitted]

50% for all the months and sites except for January. We assumed that the much smaller contributions in January may resulted partly from the longer lifetime of BCC

because of due to less wet deposition in winter., Moreover, we conducted the cluster analysis of back trajectories of air masses arriving at NJU with Hybrid Single Particle

Lagrangian Integrated Trajectory (HYSPLIT, version 4) model, and found that lessfewer air masses passed through the third modeling domain in January, as illustrated in Figure S6 in the supplement. The result thus implied more contribution from regional transport to the air quality at the site in winter compared to other seasons. We acknowledged that the multiple regression model was less effective on identifying the sources of BC in winter by constraining the emissions in southern

Jiangsu city cluster alone.Given the prevailing northerly wind directions in January, regional transport from boundary conditions accounted for the majority of contribution at two sites, the same as the result in other studies (Jia et al., 2008; Li et al., 2015; Sun et al., 2014). It thus would bring more uncertainty when estimating the top-down emissions in the multiple regression model in January. The uncertainty would be discussed in Section 3.3.

[revised manuscript text omitted]

January at PAES (especially in middle and late January, 16$^{th}$–26$^{th}$) may might result partly from the emission control policy implemented for the National Memorial Day of Nanjing Massacre Victims in December 13$^{th}$ in 2014. During the period, Nanjing was undertaking series of stringent restrictions on air pollutant emissions. For example, key petrochemical and steel industries were shut down, and all the high-pollution vehicles were forbidden to drive into the city Nanjing. Those restrictions had large impacts on emissions and thereby air quality in the following month at PAES, but have notwere been not fully considered in current emission inventories. Moreover, the bias could be enhanced under certain meteorology conditions. Meanwhile,As illustrated in Figure S89 in the supplement, higher daily average PBL height at PAES was found for periods when the simulated concentrations were relatively lower (e.g., 6$^{th}$ -7$^{th}$, 12$^{th}$-15$^{th}$ and 28$^{th}$-31$^{st}$), resulting in smaller bias between simulation and observation. we evaluated the effect of meteorology on the modeling performance at PAES, and the simulated daily average

PBL heights at PAES were illustrated in Figure S8 in the supplement. For periods when simulated concentrations were lower than other period (e.g., 6$^{th}$–7$^{th}$, 12$^{th}$-15$^{th}$

and 28$^{th}$-31$^{st}$), the simulated PBL height were higher so that it would help BC to disperse, resulting in lower simulations and smaller bias between simulations and observations. In contrast, the lower PBL height found in other periods would exaggerate the overestimation in simulated concentrations, given the elevated

批注 [zy4]: check
明确到底是什么原因导致了模拟的偏差

emissions
in JS-prior. The
seasonal variation of BC concentrations at NJU was larger than that at PAES,
suggesting bigger impact of household solid fuel use on the suburban and rural
regions. Though the model was able to capture the seasonal variability, discrepancies
between simulations and observations existed, and CTM  generally
underestimated BC concentrations at the suburban site NJU and overestimated those
at the urban site PAES. With the monthly means ranged 1.99-5.97 μg/m$^3$ at NJU, the
annual average  BC concentration (calculated as the mean of January, April, July
and October) was simulated at 3.44 μg/m$^3$, smaller than the observed 3.83 μg/m$^3$.
With the monthly means ranged 2.61-6.46 μg/m$^3$, in contrast, the annual concentration
at PAES was simulated at 3.39 μg/m$^3$, larger than the observed 2.48 μg/m$^3$. Better
correlation between observation and simulation was found at NJU, indicated by the
larger R. The annual mean NMBs were calculated at -10.16% and 36.67%, and the
NMEs were 41.15% and 72.00% at NJU and PAES, respectively. The discrepancy
suggested that JS-prior used in CTM might misrepresent the spatial pattern of
emissions. Population and economy densities were applied to allocate BC emissions,
leading to overestimation in emissions and thereby simulated concentrations in urban
areas with more population and economic activity. Besides, the model overestimated
the peak surface concentrations at both sites particularly when the contribution from
industry sector was enhanced as mentioned in Section 3.2 (e.g., January 9$^{th}$-11$^{th}$ and
April 9$^{th}$-10$^{th}$ at NJU, and April 9$^{th}$-12$^{th}$, the second half of July, and October 20$^{th}$ at
PAES).
Application of JS-posterior in CTM effectively corrected large biases between
simulations and observations at the two sites. As shown in Table 2, NMEs were
reduced for most months (all months at PAES and January and April at NJU) while
effects of applying JS-posterior in CTM varied at two sites. At PAES, the annual
average NME declined from 72.00% to 57.55% and the annual mean of BC

concentration was simulated at 2.57 μg/m³, in better agreement with the observed

2.48 μg/m³ than the simulated 3.39 μg/m³ using JS-prior. The largest reductions in

NMEs were found in April and July, from 73.18% to 42.87% and from 92.74% to

42.37%, respectively. Moreover the overestimations in peak concentrations using

JS-prior were partly corrected when JS-posterior was applied, resulting mainly from the reduced emissions from industry and transportation. Regarding the overestimation in January 16th-26th discussed above, we excluded the data points for those dates and re-compared the observation and simulation. As can be seen  in Table S6 in the supplement, The overestimation in CTM

was largely reduced  and the top-down estimate corrected the bias moderately in January at PAES. Besides the emissions, overestimation in annual BC concentration at PAES could result partly from the uncertainty in PBL modeling (Liu et al., 2018)

. As shown in Table S7 in the supplement,  the monthly PBL heights  in WRF were generally lower  than theose in actual atmosphere

, leading to the enhanced  BC concentration.

Although simulation of peak concentrations at NJU were improved as well, the annual average NME at NJU slightly increased from 41.15% to 44.16% and the annual mean of BC concentration was simulated at 2.82 μg/m³, smaller than the simulated 3.44 μg/m³ using JS-prior. Bigger bias was found in July and October at

NJU, since the reduced emission estimates in JS-posterior led to further underestimation in simulated ambient BC levels compared to JS-prior. Limitation of current multiple regression model was thus indicated that overestimation and underestimation in concentrations at different sites could hardly be corrected simultaneously without further improvement in spatial distribution of emissions.

**4 Discussions**

We selected April to evaluate the sensitivity of observation and bottom-up
emission input to top-down constraint. Observation site number, spatial
representativeness of sites, and the a  priori bottom-up inventory were changed
separately in the constraining approach, and various top-down estimates could be
derived and compared with each other.
Furthermore, we evaluated the uncertainty of the multiple
regression model, including the assumption of near linearity between emissions and
concentrations in July and October and the impact of precipitation in July. The
statistical indicators of modeling performances based on different cases are
summarized in Table 3. Details were described as below.

**4.1 The effect of observation  data application**

A major challenge in understanding the sources and distributions of BC in China
was lack of a consistent and stable measurement network with good spatiotemporal
coverage, such as the IMPROVE network in the United States (Malm et al., 1994).
Uncertainty existed in the top-down estimates in this work, as hourly measurements
on BC concentrations were only available at two sites in southern Jiangsu. Therefore,
besides JS-posterior derived from observations at both sites in April as described in
Section 3.2 (mentioned as Case 1 hereinafter), we conducted a Case 2 in which observation data at only one site (NJU)  were used in the top-down approach, to analyze the effect of the site number on emission estimates.

The scaling factors of emissions from industry, residential sources and transportation were recalculated at 0.42, 0.95 and 0.65, respectively. Compared to JS-prior in April in Table 2, the NMEs of Case 2 in Table 3

decreased from 42.31% to 32.47% and from 73.18% to 61.59% at NJU and PAES, respectively, implying the benefits of ground measurements (even available only at one site) on emission constraint. The NME in Case 2 was slightly smaller than that in

Case 1 at NJU, suggesting that application of measurement data at one single site could improve modeling performance moderately at that site. At PAES, in contrast, much larger NME was found in Case 2. Much better modeling performance in Case 1

at PAES indicated that inclusion of more measurements with better spatiotemporal coverage could constrain BC emissions at city cluster level more effectively.

It should be noted that the number of data  included in the multiple regression model was 48% of those  for the whole periods with the data screening mentioned in Section 2.2. In particular,  period lack of observation accounted for 38% of the whole month. We further analyzed the CTM performances for the  periods included in the model and those excluded from the model separately, as shown in Table

S8 in the supplement. The ~~observed concentration for the periods included in the model (2.56 μg/m$^3$)~~simulated concentration for periods included in the multiple was  smaller than the simulated in JS-prior (2.71 μg/m$^3$), leading to the reduced  emissions

then be reduced through constraining.

The constrained emissions resulted in a simulated concentration  (2.42 μg/m³  closer to the observation.  for the periods included in the multiple regression model

, and  did not increase the bias for the periods excluded from the  model. It suggested that factors other than emissions in CTM (e.g., meteorology) might contribute to underestimation for the latter

[revised manuscript text omitted]

months reflected the limitation of the top-down estimate. To evaluate the effects of
observation data on top-down estimate, two more cases in which observation data of
only one site (NJU) and observation data at both sites with their spatial
representativeness differentiated were applied to constrain the emissions, respectively.
Best CTM performance was found for the third case, indicating that inclusion of more
ground measurements with better spatiotemporal coverage in the city cluster would improve the understanding of spatial distributions of BC emissions. In addition, top-down estimates were derived from various bottom-up inventories, and the differences in emission amount, spatial distribution and CTM performance between the constrained emission estimates were significantly reduced compared to those between the bottom-up inventories. The results implied that changes in the a  priori emission input in the regression model and CTM had limited effect on the top-down estimation. Finally, the assumption of near-linearity between emissions and concentrations was justified, and the influence of wet deposition on the estimated emissions was evaluated to be moderate. This work demonstrated that top-down approach based on ground observations and CTM could capture the fast changes in BC emissions attributed to tightened pollution control policy at a city cluster scale. To further reduce uncertainty of the approach and apply the method to other regions, more ground measurements with sufficient temporal resolution would be recommended.

**Acknowledgement**

This work was sponsored by the Natural Science Foundation of China (91644220 and 41575142) and the National Key Research and Development Program of China (2017YFC0210106 and 2016YFC0201507), . We would like to acknowledge Tong Dan from Tsinghua University for national emission data (MEIC).

**References**

[revised manuscript text omitted]

Note: The criteria for the statistical significance of the model: a: $t>2$, b: Sig.$<0.05$, and c: VIF$<10$, d: the overall significance$\leq0.05$.

**Table 2. Statistical indicators for observed and simulated BC concentrations using JS-prior and JS-posterior at NJU and PAES.**

| Site | Parameter | January JS-prior | January JS-posterior | April JS-prior | April JS-posterior | July JS-prior | July JS-posterior | October JS-prior | October JS-posterior | Annual JS-prior | Annual JS-posterior |
|------|-----------|---------|-----------|---------|-----------|---------|-----------|---------|-----------|---------|-----------|
| NJU | Average SIM ($\mu g/m^3$) | 5.97 | 5.50 | 2.38 | 1.82 | 1.99 | 1.29 | 2.80 | 2.42 | 3.44 | 2.82 |
| | Average OBS ($\mu g/m^3$) | 5.44 | 5.44 | 2.69 | 2.69 | 2.65 | 2.65 | 3.96 | 3.96 | 3.83 | 3.83 |
| | NMB (%) | 8.35 | -0.08 | -16.02 | -32.40 | -23.09 | -51.32 | -29.20 | -39.01 | -10.16 | -26.43 |
| | NME (%) | 37.83 | 35.54 | 42.31 | 38.61 | 49.62 | 57.49 | 40.52 | 43.06 | 41.15 | 44.16 |
| | R | 0.67 | 0.66 | 0.34 | 0.43 | 0.36 | 0.31 | 0.42 | 0.48 | 0.67 | 0.69 |
| PAES | Average SIM ($\mu g/m^3$) | 6.46 | 5.91 | 2.98 | 1.95 | 2.61 | 1.63 | 3.19 | 2.88 | 3.39 | 2.57 |
| | Average OBS ($\mu g/m^3$) | 2.80 | 2.80 | 1.70 | 1.70 | 1.51 | 1.51 | 3.62 | 3.62 | 2.48 | 2.48 |
| | NMB (%) | 151.93 | 134.59 | 61.57 | 14.73 | 72.17 | 8.28 | -12.01 | -20.48 | 36.67 | 3.54 |
| | NME (%) | 155.53 | 139.50 | 73.18 | 42.87 | 92.74 | 42.37 | 43.10 | 40.80 | 72.00 | 57.55 |
| | R | 0.38 | 0.38 | 0.64 | 0.53 | 0.35 | 0.37 | 0.57 | 0.72 | 0.38 | 0.45 |

Note: SIM and OBS indicated the results from simulation and observation, respectively. NMB and NME were calculated using following equations (P and O indicated the results from modeling prediction and observation, respectively):

$$NMB = \frac{\sum_{i=1}^{n}(P_i - O_i)}{\sum_{i=1}^{n} O_i} \times 100\% \; ; \; . NME = \frac{\sum_{i=1}^{n}|P_i - O_i|}{\sum_{i=1}^{n} O_i} \times 100\%$$

Table 3. Statistical indicators for observed and simulated BC concentrations in different cases  at NJU and PAES (Cases 1-5 for April, and Cases 6-7 for July).

| Site | Parameter | Case1 | Case2 | Case3 | Case4 | Case5 | Case 6 | Case 7 |
|------|-----------|-------|-------|-------|-------|-------|--------|--------|
|      | Average SIM (μg/m$^3$) | 1.82 | 2.27 | 2.06 | 2.49 | 1.78 | 1.40 | 1.41 |
|      | Average OBS (μg/m$^3$) | 2.69 | 2.69 | 2.69 | 2.69 | 2.69 | 2.65 | 2.65 |
| NJU  | NMB (%) | -32.40 | -21.59 | -23.50 | -7.46 | -33.95 | -47.41 | -46.72 |
|      | NME (%) | 38.61 | 32.47 | 32.64 | 41.58 | 38.94 | 54.88 | 54.44 |
|      | R | 0.43 | 0.49 | 0.49 | 0.40 | 0.46 | 0.33 | 0.33 |
|      |   |   |   |   |   |   |   |   |
|      | Average SIM (μg/m$^3$) | 1.95 | 2.45 | 2.01 | 5.13 | 2.29 | 1.76 | 1.76 |
|      | Average OBS (μg/m$^3$) | 1.70 | 1.70 | 1.70 | 1.70 | 1.70 | 1.51 | 1.51 |
| PAES | NMB (%) | 14.73 | 49.86 | 18.02 | 201.35 | 34.71 | 16.87 | 16.65 |
|      | NME (%) | 42.87 | 61.59 | 39.62 | 201.56 | 47.73 | 44.46 | 42.71 |
|      | R | 0.53 | 0.63 | 0.66 | 0.65 | 0.59 | 0.36 | 0.39 |

Note:

Case 1 applied observations at two sites to constrain the emissions from the whole city cluster (JS-posterior); Case 2 applied observations at only one site (NJU) to constrain the whole city cluster; Case 3 applied observations at two sites to constrain emissions from different cities respectively; Case 4 applied the MEIC-prior; Case 5 applied the MEIC-posterior; Case 6 excluded the data influenced by simulated wet deposition; and Case 7 excluded the data influenced by satellite-derived accumulative precipitation.

Case 1: using observations at two sites to constrain whole city cluster (JS posterior)
Case 2: using observations at only one site (NJU) to constrain whole city cluster
Case 3: using observations at two sites to constrain different cities respectively
Case 4: using MEIC prior
Case 5: using MEIC posterior
Case 6: excluding data influenced by simulated wet deposition
Case 7: excluding data influenced by satellite-derived accumulative precipitation.

**Table 4. The scaling factors and statistical indicators from the multiple**
**regression model in Case 3.**

| Site | Sector | Scaling factor | t | Sig. | VIF |
|------|--------|----------------|------|------|------|
| NJU | Industry ($\beta_2$) | 0.42 | 1.71 | 0.09 | 2.03 |
| | Residential ($\beta_3$) | 0.95 | 2.50 | 0.01 | 2.52 |
| | Transportation ($\beta_4$) | 0.65 | 2.13 | 0.03 | 2.66 |
| PAES | Industry ($\beta_2$) | 0.19 | 3.46 | 0.00 | 1.44 |
| | Residential ($\beta_3$) | 0.36 | 1.89 | 0.06 | 1.44 |
| | Transportation($\beta_4$) | 0.65 | - | - | - |

Table 5. BC emissions from Nanjing and Suzhou–Wuxi–Changzhou–Zhenjiang city
cluster in different cases in April 2015 (Gg).

| Case | Sector | Nanjing | Suzhou–Wuxi–Changzhou–Zhenjiang | Southern Jiangsu |
|------|--------|---------|--------------------------------|------------------|
| Scenario B | Power | 0 | 0.01 | 0.01 |
| | Industry | 0.21 | 1.13 | 1.34 |
| | Residential | 0.08 | 0.24 | 0.32 |
| | Transportation | 0.12 | 0.30 | 0.42 |
| | Total | 0.41 | 1.68 | 2.09 |
| Case 1 | Power | 0 | 0.01 | 0.01 |
| | Industry | 0.05 | 0.25 | 0.30 |
| | Residential | 0.04 | 0.14 | 0.19 |
| | Transportation | 0.08 | 0.20 | 0.28 |
| | Total | 0.17 | 0.60 | 0.78 |
| Case 2 | Power | 0 | 0.01 | 0.01 |
| | Industry | 0.09 | 0.47 | 0.56 |
| | Residential | 0.07 | 0.23 | 0.30 |
| | Transportation | 0.08 | 0.20 | 0.27 |
| | Total | 0.24 | 0.91 | 1.14 |
| Case 3 | Power | 0 | 0.01 | 0.01 |
| | Industry | 0.04 | 0.47 | 0.51 |
| | Residential | 0.03 | 0.23 | 0.26 |
| | Transportation | 0.08 | 0.20 | 0.27 |
| | Total | 0.15 | 0.90 | 1.05 |

**Table 5. The scaling factors and statistical indicators from the multiple**
**regression model in Cases 6 and 7.**

Table 6. The scaling factors and statistical indicators from the multiple regression model in Case 6.

| | Month | Sector | Scaling factor | t[a] | Sig.[b] | VIF[c] | Sig.[d] |
|---|---|---|---|---|---|---|---|
| | | Industry ($\beta_2$') | 0.41 | 2.17 | 0.03 | 1.71 | |
| | January | Residential ($\beta_3$') | 1.53 | 3.48 | 0.00 | 2.29 | 0.00 |
| | | Transportation ($\beta_4$') | 0.73 | 1.65 | 0.10 | 2.66 | |
| | | Industry ($\beta_2$') | 0.24 | 0.92 | 0.36 | 1.91 | |
| | April | Residential ($\beta_3$') | 0.51 | 1.32 | 0.19 | 3.29 | 0.00 |
| | | Transportation ($\beta_4$') | 0.70 | 2.12 | 0.03 | 3.03 | |
| Case 6 | | Industry ($\beta_2$') | 0.38 | 4.43 | 0.00 | 1.43 | |
| | July | Residential ($\beta_3$') | 0.34 | 0.82 | 0.41 | 2.52 | 0.00 |
| | | Transportation ($\beta_4$') | 0.74 | 3.55 | 0.00 | 2.25 | |
| | | Industry ($\beta_2$') | 0.33 | 1.00 | 0.32 | 1.44 | |
| | October | Residential ($\beta_3$') | 1.36 | 2.61 | 0.01 | 1.86 | 0.00 |
| | | Transportation ($\beta_4$') | 0.72 | 1.89 | 0.06 | 2.02 | |
| | | Industry ($\beta_2$') | 0.38 | 2.38 | 0.02 | 1.31 | |
| Case 7 | July | Residential ($\beta_3$') | 0.31 | 0.31 | 0.75 | 2.31 | 0.00 |
| | | Transportation ($\beta_4$') | 0.75 | 1.8 | 0.07 | 1.95 | |

Note: The criteria for the statistical significance of the model: a: t>2, b: Sig.<0.05, and c: VIF<10, d: the overall significance.

**Table 76. The monthly and annual emissions by sector for southern Jiangsu 2015 in Cases 6 and 7 (unit: Gg) and the relative deviation compared to JS-posterior (RD: Case 6 or 7 – JS-posterior)/ JS-posterior, %).**

| | January | | April | | July | | October | | Annual | | July | |
|---|---|---|---|---|---|---|---|---|---|---|---|---|
| | Case 6 | RD | Case 6 | RD | Case 6 | RD | Case 6 | RD | Case 6 | RD | Case 7 | RD |
| Power | 0.0 | 0.0% | 0.0 | 0.0% | 0.0 | 0.0% | 0.0 | 0.0% | 0.0 | 0.0% | 0.0 | 0.0 |
| Industry | 0.6 | -2.4% | 0.3 | 9.9% | 0.6 | 9.2% | 0.5 | -0.3% | 6.0 | 3.1% | 0.5 | 9.5 |
| Residential | 0.5 | 16.7% | 0.2 | -13.1% | 0.1 | -13.7% | 0.4 | -10.2% | 3.6 | -0.6% | 0.1 | -20.6 |
| Transportation | 0.3 | -8.2% | 0.3 | 4.3% | 0.4 | 34.4% | 0.3 | -3.0% | 3.9 | 5.4% | 0.4 | 36.4 |
| Sum | 1.4 | 2.4% | 0.8 | 2.3% | 1.1 | 13.6% | 1.2 | -4.2% | 13.5 | 2.6% | 1.0 | 13.4 |

**Table 8. Statistical indicators for the observed and simulated BC concentrations**
**in July 2015 at NJU and PAES in Case 6 and Case 7.**

| — | Parameter | Case 6 | Case 7 |
|---|---|---|---|
| NJU | Average SIM ($\mu g/m^3$) | 1.40 | 1.41 |
| | Average OBS ($\mu g/m^3$) | 2.65 | 2.65 |
| | NMB (%) | -47.41 | -46.72 |
| | NME (%) | 54.88 | 54.44 |
| | R | 0.33 | 0.33 |
| PAES | Average SIM ($\mu g/m^3$) | 1.76 | 1.76 |
| | Average OBS ($\mu g/m^3$) | 1.51 | 1.51 |
| | NMB (%) | 16.87 | 16.65 |
| | NME (%) | 44.46 | 42.71 |
| | R | 0.36 | 0.39 |

Note: SIM and OBS indicated the results from simulation and observation,
respectively. NMB and NME were calculated using following equations (P and O
indicated the results from modeling prediction and observation, respectively):

$$NMB = \frac{\sum_{i=1}^{n}(P_i - O_i)}{\sum_{i=1}^{n}O_i} \times 100\% \qquad NME = \frac{\sum_{i=1}^{n}|P_i - O_i|}{\sum_{i=1}^{n}O_i} \times 100\%$$
;

Table 9. The scaling factors and statistical indicators from the multiple
regression model in Case 7. The emissions by sector for southern Jiangsu 2015
July in Case 7 (unit: Gg) and the relative deviations (RD) compared to Case 1
(RD: Case 7 - Case 1)/Case 1) are also shown in table.

| Sector | Scaling factor | $t^a$ | Sig.$^b$ | VIF$^c$ | Sig.$^d$ | Emissions | RD |
|---|---|---|---|---|---|---|---|
| Power | | | | | | 0.0 | 0.0% |
| Industry ($\beta_2$') | 0.38 | 2.38 | 0.02 | 1.31 | | 0.5 | 9.5% |
| Residential ($\beta_3$') | 0.31 | 0.31 | 0.75 | 2.31 | 0.00 | 0.1 | -20.6% |
| Transportation ($\beta_4$') | 0.75 | 1.8 | 0.07 | 1.95 | | 0.4 | 36.4% |
| Sum | | | | | | 1.0 | 13.4% |

Note: The criteria for the statistical significance of the model: a: t>2, b: Sig.<0.05, and c: VIF<10, d: the overall significance.

**Figure 1**

[Figure]

Figure 2

[Figure]

**Figure 2**

[Figure]

**Figure 3Figure 3**

[Figure]

(a)

[Figure]

(b)

 Figure 4

[Figure]

(a)

(b)

(c)

[Figure]

(d)

[Figure]

(a)

(b)

(c)

[Figure]

(d)

**Figure 56**

[Figure]

**Figure 67**

[Figure]

(a)

[Figure]

(b)

 **Figure 78**

[Figure]

(a)

[Figure]

(b)

Figure 9

[Figure]

(a)

(b)

[Figure]

(c)

(d)

[Figure]

(a)

[Figure]

(b)

[Figure]

(c)

[Figure]

(d)

**Figure 811**

[Figure]

(a)

[Figure]

(b)

**Figure 9**

[Figure]

(a)

[Figure]

(b)